# Rate-Distortion Optimization Guided Autoencoder for Generative Approach

## Abstract

In the generative model approach of machine learning, it is essential to acquire an accurate probabilistic model and compress the dimension of data for easy treatment. However, in the conventional deep-autoencoder based generative model such as VAE, the probability of the real space cannot be obtained correctly from that of in the latent space, because the scaling between both spaces is not controlled. This has also been an obstacle to quantifying the impact of the variation of latent variables on data. In this paper, we propose Rate-Distortion Optimization guided autoencoder, in which the Jacobi matrix from real space to latent space has orthonormality. It is proved theoretically and experimentally that (i) the probability distribution of the latent space obtained by this model is proportional to the probability distribution of the real space because Jacobian between two spaces is constant; (ii) our model behaves as non-linear PCA, where energy of acquired latent space is concentrated on several principal components and the influence of each component can be evaluated quantitatively. Furthermore, to verify the usefulness on the practical application, we evaluate its performance in unsupervised anomaly detection and it outperforms current state-of-the-art methods.

## 1 Introduction

Capturing the inherent features of a dataset from high-dimensional and complex data is an essential issue in machine learning. Generative model approach learns the probability distribution of data, aiming at data generation by probabilistic sampling, unsupervised/weakly supervised learning, and acquiring meta-prior (general assumptions about how data can be summarized naturally, such as disentangle, clustering, and hierarchical structure (Bengio et al., 2013; Tschannen et al., 2019)). It is generally difficult to directly estimate a probability density function(PDF) $Px(\boldsymbol{x})$ of real data $\boldsymbol{x}$. Accordingly, one promising approach is to map to the latent space $\boldsymbol{z}$ with reduced dimension and capture PDF $Pz(\boldsymbol{z})$. In recent years, deep autoencoder based methods have made it possible to compress dimensions and derive latent variables. While there is remarkable progress in these areas (van den Oord et al., 2017; Kingma et al., 2014; Jiang et al., 2016), the relation between $\boldsymbol{x}$ and $\boldsymbol{z}$ in the current deep generative models is still not clear.

Variational autoencoder (VAE) (P.Kingma & Welling, 2014) is one of the most successful generative models for capturing latent representation. In VAE, lower bound of log-likelihood of $Px(\boldsymbol{x})$ is introduced as ELBO. Then latent variable is obtained by maximizing ELBO. In order to maximize ELBO, various methods (Alemi et al., 2018; Zhao et al., 2019; Brekelmans et al., 2019) have been proposed. However, many previous works did not care about the value of Jacobian between two spaces, despite the fact that the ratio between $Pz(\boldsymbol{z})$ and $Px(\boldsymbol{x})$ is equal to the Jacobian. Even in models that provide more flexible estimation (Johnson et al., 2016; Liao et al., 2018; Zong et al., 2018), this point is overlooked.

Here, when we turn our sight toward acquiring meta-prior, it is straightforward to evaluate the quantitative influence of each latent variable on the given metrics between data $x_1$ and $x_2$. To do so, the scale of the latent variable should be appropriately controlled so that the changes in latent variables is proportional to the changes of given metrics in data space. In addition, this scaling should be adjusted according to the definition of the metrics. For instance, with respect to image quality metrics, different meta-prior should be derived from mean square error (MSE) and the structural similarity (SSIM). Considering the mechanism of principal component analysis (PCA) would be one of the

directions to solve it. In PCA, PCA components are derived by optimal orthonormal transformation, then the importance of each component can be identified quantitatively by their variance. It should be noted that orthogonality is not enough for quantitative analysis, but both orthogonality and normalized scaling, namely, orthonormality is required. This fact implies that, if Jacobi matrix of autoencoder has orthonormality, the characteristics of acquired latent space can be evaluated quantitatively. To deal with this, we propose RaDOGAGA (Rate-Distortion Optimization Guided Autoencoder for Generative Approach), based on the rate-distortion optimization (RDO) which has been widely used in image compression with orthonormal transform coding (Sullivan & Wiegand (1998)). In this paper, we show the effect of RaDOGAGA in the following steps.

(1) We prove that RaDOGAGA has the following property theoretically and experimentally.

- Jacobi matrix between real space and latent space leads to be constant-scaled orthonormal. So the response of the minute change of $z$ to the real space data $x$ is constant at any $z$.
- Because of constant Jacobian (or pseudo-Jacobian), $Px(x)$ and $Pz(z)$ are almost proportional. Therefore, $Px(x)$ can be estimated, by directly maximizing log-likelihood of parametric PDF $Pz_\psi(z)$ in reduced-dimensional space, without considering ELBO.
- When univariate independent distribution is used to estimate $Pz(z)$ parametrically, it behaves as "continuous PCA" where energy is concentrated on several principal components.

(2) Thanks to this property, RaDOGAGA achieve the state-of-the-art performance in anomaly detection task with four public datasets, where probability density estimation is important.

(3) We show that our approach can directly evaluate how the $z$ impact on the given metrics in real space. This feature is promising to further interpretation of latent variables.

## 2 RELATED WORK

**Flow based model:** Flow based generative models generates astonishing quality of image (Kingma & Dhariwal, 2018; Dinh et al., 2014). Flow mechanism explicitly takes Jacobian of $x$ and $z$ into account. The transformation function $z = f(x)$ is learned, calculating and preserving Jacobian of $x$ and $z$. Unlike ordinary autoencoder, which reverse $z$ to $x$ with function $g(\cdot)$ different from $f(\cdot)$, inverse function transforms $z$ as $x = f^{-1}(z)$. Since the model preserves Jacobian, $Pz(x)$ can be estimated by maximizing log likelihood of $Pz(z)$ without considering ELBO. Although, in this approach, $f(\cdot)$ need to be bijection. Because of this limitation, it is difficult to fully utilize the flexibility of neural networks.

**Interpretation of latent variables:** While it is expected to acquire meta-prior by deep autoencoder, interpreting latent variables is still challenging. In recent years, research aiming to acquire disentangled latent variables, which encourages them to be independent, is flourishing (Lopez et al., 2018; Chen et al., 2018; Kim & Mnih, 2018; Chen et al., 2016). With these methods, qualitative effects for disentanglement can be seen. For example, when a certain latent variable is displaced, image changes corresponding to specific attributes (size, color, etc.). Some works also propose quantitative metrics for meta-prior. In beta-VAE (Higgins et al., 2017), the metric evaluates the independence of latent variables by solving the classification task. Actually, the Jacobi matrix of VAE is orthogonal (Rolínek et al., 2019), which enables to make latent variables disentangled implicitly. However, orthonormality is not supported and it is difficult to define the metric which evaluates the effect of latent variables to the metrics between data directly.

**Deep image compression with rate-distortion optimization:** Rate-distortion (RD) theory (Berger, 1971) is a part of Shannon's information theory for lossy compression which formulates the optimal condition between information rate and distortion. Then RD theory for Gaussian source with memory has been further extended to transform coding (Goyal, 2001) for image and audio lossy compression where orthonormal transforms such as Karhunen-Loève transform (KLT) (Rao & Yip (2000)) and discrete cosine transform (DCT) are used for decorrelation. Furthermore, rate-distortion optimization (RDO) has been widely used in image compression (Sullivan & Wiegand, 1998). In RDO, a cost $L = R + \lambda \cdot D$ is minimized at given Lagrange parameter $\lambda$ to realize the best trade-off between rate $R$ and distortion $D$ such that $L = R + \lambda \cdot D$ becomes a tangent line of true Rate-Distortion curve. Recently, deep learning based image compression (Ballé et al., 2018; Zhou et al., 2019) has been proposed. In these works, instead of orthonormal transform with L2 norm metrics in the conventional lossy compression method, a deep autoencoder is trained with flexible metrics

such as SSIM for RDO. To understand the success of these works, we prove theoretically that RDO guides a Jacobi matrix of deep autoencoder to orthonormal in the space depending on the metrics. In the next section, we explain the idea of RDO guided autoencoder and its relationship with VAE.

## 3 OVERVIEW OF RATE-DISTORTION OPTIMIZATION GUIDED APPROACH

### 3.1 DERIVATION FROM RATE-DISTORTION OPTIMAZATION IN TRANSFORM CODING

Figure 1 shows the overview of our idea inspired by Rate-Distortion optimization of transform coding. In the transform coding, the optimal method to encode data with Gaussian distribution is as follows (Goyal, 2001). At first, the data are transformed deterministically to decorrelated data using the orthonormal transform such as KLT and DCT. Then these decorrelated data are quantized stochastically with uniform quantizer for all channels such that quantization noise for each channel is equal. At last optimal entropy encoding is applied to quantized data where the rate can be calculated by the logarithm of symbol's estimated probability. From this fact, we have an intuition that the Jacobi matrix of autoencoder becomes orthonormal if the data were compressed based on Rate-Distortion optimization with uniform quantized noise and parametric distribution of latent variables. Inspired by this, we propose autoencoder which scales latent variables according to the definition of metrics of data, without limitation of the transformation function as in Flow models. Thanks to this feature, our scheme can estimate $Pz(\boldsymbol{z})$ quantitatively, which is suitable for clustering and anomaly detection. Furthermore, in the case factorized distribution is used for $Pz(\boldsymbol{z})$, our model behaves as continuous PCA. This property is considered to promote the interpretation of latent variables.

### 3.2 RELATIONSHIP WITH VAE

There is a number of VAE studies that taking rate-distortion trade-off into account. In VAEs, instead of maximizing log-likelihood of $P(x)$ directly, maximizing ELBO. In beta-VAE (Higgins et al., 2017), Objective function $L_{VAE}$ is described as $L_{VAE} = L_{rec} - \beta \cdot L_{kl}$. $L_{kl}$ is KL divergence between encoder output and prior distribution, usually Gaussian distribution. By changing $\beta$, the rate-distortion trade-off at desirable rate can be realized as discussed in (Alemi et al., 2018).

When the relation between beta-VAE and RDO in image compression is considered, both are analogous to each other. That is, $\beta^{-1}$ is corresponding to $\lambda$, $-L_{kl}$ is corresponding to a rate $R$, and $L_{rec}$ is corresponding to a distortion $D$. However, the probability distribution models of latent variables are quite different. VAE uses a fixed prior distribution. This causes a non-linear scaling relationship between real data and latent variables.

Figure 2 shows the conditions to achieve Rate-Distortion optimization in both VAE and RDO guided autoencoder. In VAE, precise control of noise distribution is required for each data and channel as suggested in Brekelmans et al. (2019) in order to compensate for the nonlinearities between the data space and the latent space. Jacobi matrix becomes orthogonal as proved in Rolínek et al. (2019). However, Jacobian is not constant because of non-linear scaling. In RDO guided autoencoder, uniform noises are added to all channels. Instead, parametric probability distribution should be used in order to make the scaling linear between the data space and the latent space. As a result, the Jacobi matrix becomes orthonormal because both orthogonality and scaling normalization are simultaneously achieved. As discussed above, the precise noise control in VAE and distribution parameter optimization in RDO guided autoencoder are essentially the same. Accordingly, complexities in both methods are estimated to be at the same degree.

## 4 METHOD AND THEORETICAL PROPERTY

### 4.1 METHOD

Our model is based on the Rate-Distortion optimization of the autoencoder for the hyperprior proposed in Ballé et al. (2018) for image compression with some modification. The cost function consists of (i) reconstruction error $D$ between input data and decoder output with noise to latent variable and (ii) Rate $R$ of latent variable. This is analogous to beta-VAE where $\lambda = \beta^{-1}$.

$$L = R + \lambda \cdot D \tag{1}$$

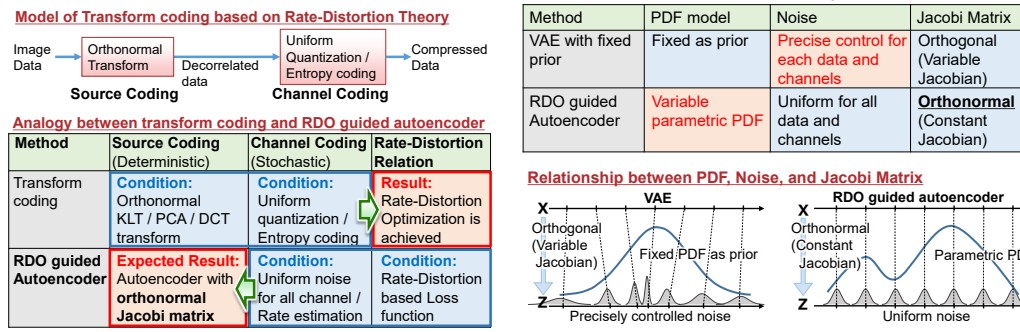

Figure 1: Overview of our idea based on RDO.

Figure 2: The condition to optimize rate-distortion in VAE and our models

The specific method is described as follows. First, let $\boldsymbol{x}$ be $M$-dimensional domain data($\boldsymbol{x} \in \boldsymbol{R}^M$) and $Px(\boldsymbol{x})$ the probability of $\boldsymbol{x}$. Then $\boldsymbol{x}$ is converted to $N$-dimensional latent space $\boldsymbol{z} \in \boldsymbol{R}^N$ by encoder. Let $f_\theta(\boldsymbol{x})$, $g_\phi(\boldsymbol{z})$, and $Pz_\psi(\boldsymbol{z})$ be parametric encoder, decoder, and probability distribution function of latent space with parameters $\theta$, $\phi$, and $\psi$. It is noted that encoder and decoder are deterministic unlike VAE. Then latent variable $\boldsymbol{z}$ and decoded data $\hat{\boldsymbol{x}}$ are generated as bellow:

$$\boldsymbol{z} = f_\theta(\boldsymbol{x}) \qquad \hat{\boldsymbol{x}} = g_\phi(\boldsymbol{z}) \tag{2}$$

Let $\boldsymbol{\epsilon} \in \boldsymbol{R}^N$ be noise with each dimension being independent with an average of 0 and a equal variance of $\sigma^2$ to emulate uniform quantization.

$$\boldsymbol{\epsilon} = (\epsilon_1, \epsilon_2, ..\epsilon_N), \ E\left[\epsilon_i\right] = 0, \ E\left[\epsilon_i \cdot \epsilon_j\right] = \delta_{ij} \cdot \sigma^2 \tag{3}$$

Then, given the sum of latent variable $\boldsymbol{z}$ and noise $\boldsymbol{\epsilon}$, the decoder output $\breve{\boldsymbol{x}}$ is obtained. This is analogous to the stochastic sampling and decoding procedure in VAE.

$$\breve{\boldsymbol{x}} = g_\phi(\boldsymbol{z} + \boldsymbol{\epsilon}) \tag{4}$$

Here, the cost function is defined by Eq. (5) with some modifications from Eq. (1).

$$L = -\log(Pz_\psi(\boldsymbol{z})) + \lambda_1 \cdot h\left(D\left(\boldsymbol{x}, \hat{\boldsymbol{x}}\right)\right) + \lambda_2 \cdot D\left(\hat{\boldsymbol{x}}, \breve{\boldsymbol{x}}\right) \tag{5}$$

In this equation, the first term is corresponding to the estimated rate of the latent variables. In Ballé et al. (2018), a rate for quantized symbol is calculated by $-\int_{-0.5}^{0.5} \log(Pz_\psi(\boldsymbol{z}))d\boldsymbol{z}$. In our model, this is replaced by $-\log(Pz_\psi(\boldsymbol{z}))$ for simplicity. To model $Pz_\psi(\boldsymbol{z})$, factorized probability model or Gaussian mixture model (GMM) can be used for instance. $D(\boldsymbol{x}_1, \boldsymbol{x}_2)$ in the second and the third term is a metrics function between $\boldsymbol{x}_1$ and $\boldsymbol{x}_2$. Actually, the second and the third term is decomposition of $D(\boldsymbol{x}, g_\phi(f_\theta(\boldsymbol{x})+\boldsymbol{\epsilon})) \sim D(\boldsymbol{x}, g_\phi(f_\theta(\boldsymbol{x})))+D(g_\phi(f_\theta(\boldsymbol{x})), g_\phi(f_\theta(\boldsymbol{x})+\boldsymbol{\epsilon}))$ as shown in Rolínek et al. (2019). The second term purely calculate reconstruction loss and the third term only influences the scaling of latent space. Accordingly, $\lambda_1$ controls the degree of reconstruction, and $\lambda_2(\sim \beta^{-1}$ of beta-VAE) controls a scaling between the data space and the latent space. By this decomposition, we can adjust balance between reconstruction loss and other losses flexibly and easily lead better performance. $h(\cdot)$ in the second term of Eq. (5) is a monotonically increasing function. According to the choice of $h(\cdot)$, the degree of reconstruction accuracy can be controlled. Roughly speaking, when $h(d) = d$ is chosen, the training much focus on fitting the distribution and entropy. On the other hand, $h(d) = \log(d)$ encourages better reconstruction, since the curve of loss is steep around 0. The effect of $h(d)$ is further discussed in Appendix D. Then, Eq. (5) is averaged according to distributions, $\boldsymbol{x} \sim Px(\boldsymbol{x})$ and $\boldsymbol{\epsilon} \sim P(\epsilon)$. For metrics function $D(\cdot, \cdot)$, a variety of function is applicable such as MSE, SSIM, and so on. As long as the function can be approximated by the following quadratic form in the neighborhood of x, the property of the model is hold

$$D(\boldsymbol{x}, \boldsymbol{x} + \boldsymbol{\delta x}) \sim {}^t\boldsymbol{\delta x} \ \boldsymbol{A}(\boldsymbol{x}) \ \boldsymbol{\delta x} = \|\boldsymbol{L}(\boldsymbol{x}) \ \boldsymbol{\delta x}\|^2 \tag{6}$$

Here, $\boldsymbol{\delta x}$ is arbitrary micro variation of $\boldsymbol{x}$, $\boldsymbol{A}(\boldsymbol{x})$ is $M \times M$ positive definite matrix depending on $\boldsymbol{x}$, and $\boldsymbol{L}(\boldsymbol{x})$ is Cholesky decomposition of $\boldsymbol{A}(\boldsymbol{x})$. For instance, when $D(\cdot, \cdot)$ is square of L2 norm as in Eq. (7), $\boldsymbol{A}(\boldsymbol{x})$ and $\boldsymbol{L}(\boldsymbol{x})$ are Identity matrices.

$$D(\boldsymbol{x}_1, \boldsymbol{x}_2) = \|\boldsymbol{x}_1 - \boldsymbol{x}_2\|^2 \tag{7}$$

In the case of SSIM (Wang et al., 2001) metric which is close to subjective image quality, a cost $(1 - SSIM)$ can be also approximated in a quadratic form with positive definite matrix. This is explained in Appendix C. By deriving parameters that minimize this value, the encoder, decoder, and probability distribution of the latent space are trained as Eq. (8).

$$\theta, \phi, \psi = \arg\min_{\theta,\phi,\psi}(E_{\boldsymbol{x}\sim Px(\boldsymbol{x}),\ \boldsymbol{\epsilon}\sim P(\boldsymbol{\epsilon})}[\ L\ ]) \tag{8}$$

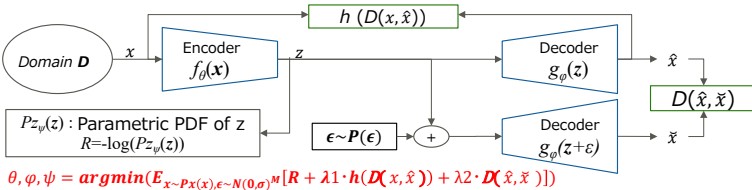

Figure 3: Architecture of RaDOGAGA

## 4.2 THEORETICAL PROPERTY OF THE MODEL

In this section, we show the theoretical property of the model. First of all, we provide a rough sketch of the theory. Let $J = |\mathrm{d}\boldsymbol{x}/\mathrm{d}\boldsymbol{z}|$ be a Jacobian between the data space $\boldsymbol{x}$ and the latent space. Here, $R$ and $\lambda D$ in Eq. (1) can be approximated by $-\log Px(\mathrm{x}) - \log(J)$ and $C_{\sigma,\lambda}J^2$ respectively where $C_{\sigma,\lambda}$ is a constant depending on $\sigma$ and $\lambda$. As a result, the Jacobi matrix with $J = 1/\sqrt{2C_{\sigma,\lambda}}$ minimizes Eq. (1). Because Jacobi matrix is orthogonal (Rolínek et al., 2019) and the Jacobian is constant, the Jacobi matrix becomes orthonormal. A full proof is described in Appendix A. In this section, we mainly explain important points. Let $\boldsymbol{x}_D$ be rescaled space of $\boldsymbol{x}$ according to the metrics $D(\cdot,\cdot)$, where $\mathrm{d}\boldsymbol{x}$ and $\mathrm{d}\boldsymbol{x}_D$ are differentials and $\boldsymbol{L}(\boldsymbol{x})$ is a Jacobi matrix between two spaces.

$$\mathrm{d}\boldsymbol{x}_D = \boldsymbol{L}(\boldsymbol{x})\mathrm{d}\boldsymbol{x} \tag{9}$$

When $D(\cdot,\cdot)$ is square of L2 norm, both spaces of $\boldsymbol{x}_D$ and $\boldsymbol{x}$ are equivalent. As described above, each column of the Jacobian matrix of latent space $\boldsymbol{z}$ and rescaled space $\boldsymbol{x}_D$ are constant multiple of orthonormal basis regardless of the value of $\boldsymbol{z}$ and $\boldsymbol{x}$ after training based on Eqs. (5) and (8). Here, $\delta_{ij}$ denotes Kronecker delta and matrix ${}^t\boldsymbol{T}$ denotes a transpose of $\boldsymbol{T}$.

$$
{}^t\left(\boldsymbol{L}(\boldsymbol{x})\frac{\partial \boldsymbol{x}}{\partial z_i}\right)\left(\boldsymbol{L}(\boldsymbol{x})\frac{\partial \boldsymbol{x}}{\partial z_j}\right) = {}^t\left(\frac{\partial \boldsymbol{x}_D}{\partial z_i}\right)\left(\frac{\partial \boldsymbol{x}_D}{\partial z_j}\right) = \frac{1}{2\lambda_2\sigma^2}\cdot\delta_{ij} \tag{10}
$$

From the orthonormal property, our model has several features. The first feature is that The distance derivatives of both real and latent spaces has a linear relationship. Therefore, the metrics between $\boldsymbol{x}$ and $\boldsymbol{x} + \mathrm{d}\boldsymbol{x}$ is proportional to $\|\mathrm{d}\boldsymbol{z}\|^2$ as derived by Eqs. (6), (9) and (10) where $\mathrm{d}\boldsymbol{z}$ is a differential.

$$
D(\boldsymbol{x},\boldsymbol{x}+\mathrm{d}\boldsymbol{x}) = \|\boldsymbol{L}(\boldsymbol{x})\mathrm{d}\boldsymbol{x}\|^2 = \|\mathrm{d}\boldsymbol{x}_D\|^2 = \left\|\sum_{i=1}^{N}\frac{\partial \boldsymbol{x}_D}{\partial z_i}\cdot\mathrm{d}z_i\right\|^2 = \frac{1}{2\lambda_2\sigma^2}\cdot\|\mathrm{d}\boldsymbol{z}\|^2 \tag{11}
$$

$$
D(\boldsymbol{x},\boldsymbol{x}+\mathrm{d}\boldsymbol{x})/\|\mathrm{d}\boldsymbol{z}\|^2 = \|\mathrm{d}\boldsymbol{x}_D\|^2/\|\mathrm{d}\boldsymbol{z}\|^2 = const. \tag{12}
$$

This equation also implies that the L2 distance of two data in the latent space is proportional to the distance between corresponding data in the real space which is defined by the integration of the root of metrics derivative.

The second feature is constant Jacobian. In case of $M = N$, the Jacobian matrix $\mathrm{d}\boldsymbol{x}_D/\mathrm{d}\boldsymbol{z}$ is a square matrix, and each column is the same as $1/(\sqrt{2\lambda_2}\cdot\sigma)$ times orthonormal basis. For this reason, the Jacobian is a constant regardless of the value of $\boldsymbol{z}$ as shown below:

$$
\left|\frac{\mathrm{d}\boldsymbol{x}_D}{\mathrm{d}\boldsymbol{z}}\right| = \left(\frac{1}{2\lambda_2\cdot\sigma^2}\right)^{(N/2)} \tag{13}
$$

In this case, the probability distribution of $\boldsymbol{x}_D$ and $\boldsymbol{z}$ is proportional because of the constant Jacobian. For the case of $M > N$, we assume the situation where most energy is efficiently and effectively mapped to N-dimensional latent space. Then the product of the singular values of SVD for a Jacobi

matrix can be regarded as a pseudo Jacobian between the real space and the latent space. Since all of the N singular values are $1/(\sqrt{2\lambda_2} \cdot \sigma)$, the pseudo Jacobian is also a constant.

The third feature is proportional probability density between data space and latent space. Because Jacobian or pseudo Jacobian $J$ is constant, the following equation holds. $P\boldsymbol{x}_D(\boldsymbol{x}_D)$ and $Pz(\boldsymbol{z})$ are are true PDFs for each space $\boldsymbol{x}_D$ and $\boldsymbol{z}$ respectively.

$$Pz(\boldsymbol{z}) \sim J \cdot P\boldsymbol{x}_D(\boldsymbol{x}_D) \propto P\boldsymbol{x}_D(\boldsymbol{x}_D) \tag{14}$$

The forth is that the log-likelihood of PDF of the domain data can be maximized without considering ELBO. This is revealed by the transformation of Equation (5). Let $\hat{P}\boldsymbol{x}_D(\boldsymbol{x}_D)$ be estimated probability of $\boldsymbol{x}_D$. Because the Jacobian J is constant, $\hat{P}\boldsymbol{x}_D(\boldsymbol{x}_D)$ can be approximated by $J^{-1} \cdot Pz_\psi(\boldsymbol{z})$. Accordingly, the average of the first term in Equation (5) by $\boldsymbol{x} \sim P\boldsymbol{x}(\boldsymbol{x})$ can be transformed as follows.

$$E_{\boldsymbol{z} \sim Pz(\boldsymbol{z}))}[-\log(Pz_\psi(\boldsymbol{z}))] \simeq -E_{\boldsymbol{x}_D \sim Px(\boldsymbol{x}_D))}[\log(\hat{P}\boldsymbol{x}_D(\boldsymbol{x}_D)] - \log(J) \tag{15}$$

Consequently, minimization of Equation (15) is equivalent to the log-likelihood maximization of $\hat{P}\boldsymbol{x}_D(\boldsymbol{x}_D)$.

The fifth is "continuous PCA" feature when the following factorized probability model is used.

$$Pz_\psi(\boldsymbol{z}) = \prod_{i=1}^{N} Pz_{i\psi}(Z_i) \tag{16}$$

As shown in FactorVAE (Kim & Mnih, 2018), the use of factorized model minimizes mutual information among latent variables and encourages disentangling. Combining with the first character of our model described before, the variance values of each latent variable show the quantitative impact on data space like linear PCA. The detailed derivation is explained in Appendix B. .

## 5 EXPERIMENT

### 5.1 PROBABILITY DENSITY ESTIMATION WITH TOY DATA

In this section, we describe our experiment using toy data to demonstrate whether the probability density of the input data $Px(\boldsymbol{x})$ and that of estimated in the latent space $Pz(\boldsymbol{z})$ are proportional to each other as in theory. First, we sample data $\boldsymbol{s} = (s_1, s_2...s_{10,000})$ from three-dimensional Gaussian distribution consists of three-mixture-components with mixture weight $\boldsymbol{\pi} = (1/3, 1/3, 1/3)$, mean $\boldsymbol{\mu}_k = (\mu_{k1}, \mu_{k2}, \mu_{k3})$, and covariance $\boldsymbol{\Sigma}_k = diag(\sigma_{k1}, \sigma_{k2}, \sigma_{k3})$. $k$ is the index for mixture component. Then, we scatter $\boldsymbol{s}$ with uniform random noise $\boldsymbol{u} \in R^{3 \times 16}$, $u_{dm} \sim U_d\left(-\frac{1}{2}, \frac{1}{2}\right)$, where $d$ and $m$ are index for dimension of sampled data and scattered data. The $U_d$s are uncorrelated with each other. We produce 16-dimensional input data $\boldsymbol{x}$ with a sample number of 10,000 in the end.

$$\boldsymbol{x} = \sum_{d=1}^{3} \boldsymbol{u}_d s_d \tag{17}$$

The appearance probability of the input data $Px(\boldsymbol{x})$ is equivalent to the generation probability of $s$.

### 5.1.1 CONFIGURATION

In the experiment, we estimate the $Pz_\psi(\boldsymbol{z})$ using GMM with parameter $\psi$ as in DAGMM (Zong et al., 2018). Instead of EM algorithm, GMM parameters are learned using Estimation Network (EN), which consists of multi-layer neural network. When the GMM has $K$ mixture-components and $L$ is the size of batch samples, EN outputs the mixture-components membership prediction as $K$-dimensional vector $\widehat{\gamma}$ as follows:

$$\boldsymbol{p} = EN(\boldsymbol{z}; \psi), \widehat{\gamma} = softmax(\boldsymbol{p}) \tag{18}$$

K-th mixture weight $\widehat{\phi}_k$, mean $\widehat{\mu}_k$, covariance $\widehat{\Sigma}_k$, and entropy $R$ of $\boldsymbol{z}$ are further calculated by Eqs. (19) and (20).

$$\widehat{\pi}_k = \sum_{l=1}^{L} \widehat{\gamma}_{lk}/L, \quad \widehat{\mu}_k = \sum_{l=1}^{L} \widehat{\gamma}_{lk}\boldsymbol{z}_l / \sum_{l=1}^{L} \widehat{\gamma}_{lk}, \quad \widehat{\boldsymbol{\Sigma}}_k = \sum_{l=1}^{L} \widehat{\gamma}_{lk}(\boldsymbol{z}_l - \widehat{\mu}_k)(\boldsymbol{z}_l - \widehat{\mu}_k)^T / \sum_{l=1}^{L} \widehat{\gamma}_{lk} \tag{19}$$

$$R = -\log\left(\sum_{k=1}^{L} \widehat{\pi}_k \Big/ \sqrt{\left|2\pi\widehat{\mathbf{\Sigma}}_k\right|} \cdot \exp\left(-\frac{1}{2}(\mathbf{z} - \widehat{\mu}_k)^T \widehat{\mathbf{\Sigma}}_k^{-1}(\mathbf{z} - \widehat{\mu}_k)\right)\right) \tag{20}$$

Overall network is trained by Eqs. (5) and (8). In this experiment, we set $D(x_1, x_2)$ as square error $\| x_1 - x_2 \|^2$, and test two types of $h(\cdot)$, $h(d) = d$ and $h(d) = \log(d)$. We denote models trained with these $h(\cdot)$ as RaDOGAGA(d) and RaDOGAGA(log(d)). As a comparison method, DAGMM is used. DAGMM also consists of encoder, decoder, and EN. In DAGMM, to avoid falling into the trivial solution that entropy is minimized when the diagonal component of the covariance matrix is 0, the inverse of the diagonal component is added to the cost function as Eq. (21):

$$L = \|\mathbf{x} - \hat{\mathbf{x}}\|^2 + \lambda_1 \cdot (-\log(Pz_\psi(\mathbf{z}))) + \lambda_2 P(\widehat{\mathbf{\Sigma}}), \qquad P(\widehat{\mathbf{\Sigma}}) = \sum_{k=1}^{K}\sum_{i=1}^{N} \widehat{\mathbf{\Sigma}}_{kii}^{-1} \tag{21}$$

The only differences between our model and DAGMM are (i) RDO mechanism is introduced (ii) the third regularization term in Eq. (21) is eliminated, since our model adjust latent scale appropriately so as not to falling into trivial solution. Thus, model complexity such as the number or parameter is the same. For both RaDOGAGA and DAGMM, the autoencoder part is constructed with fully connected (FC) layers with sizes of 64, 32, 16, 3, 16, 32, and 64. For all FC layers except for the last of the encoder and the decoder, we attach $tanh$ as the activation function. The EN part is also constructed with FC layer with a size of 10, 3. For the first layer, we attach the $tanh$ as activation function and dropout (ratio=0.5). For the last one, softmax is attached. $(\lambda_1, \lambda_2)$ is set as $(1 \times 10^{-4}, 1 \times 10^{-9})$, $(1 \times 10^6, 1 \times 10^6)$ and $(1 \times 10^3, 1 \times 10^4)$ for DAGMM, RaDOGAGA(d) and RaDOGAGA(log(d)) respectively. Optimization is done by Adam optimizer (Kingma & Ba, 2014) with learning rate $1 \times 10^{-4}$ for all model. We set $\sigma^2$ as $1/12$.

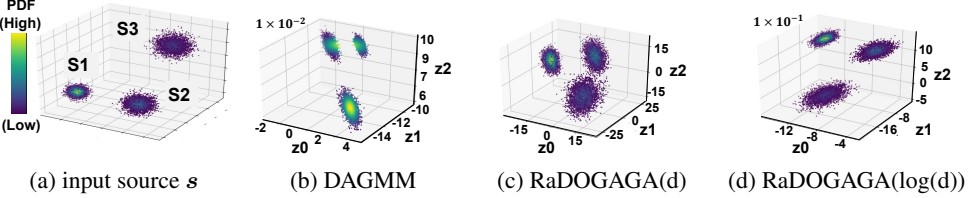

(a) input source $s$    (b) DAGMM    (c) RaDOGAGA(d)    (d) RaDOGAGA(log(d))

Figure 4: Plot of source of input data $s$ and latent variable $z$. Normalized PDF value corresponds to a color bar located left of (a). Even though both DAGMM and RaDOGAGA capture three mixture components, the tendency of PDF distribution in DAGMM looks different from input data source. Points with high PDF are not concentrated on the center of the clustering especially in upper two.

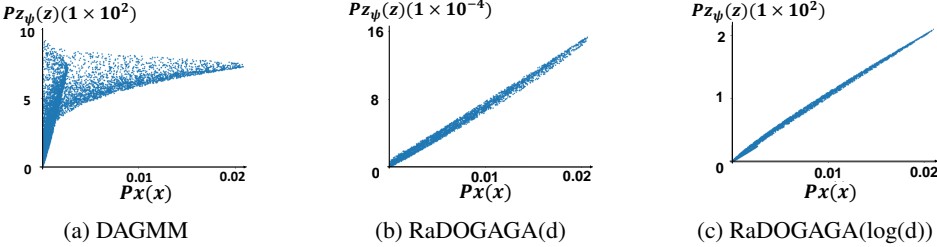

(a) DAGMM      (b) RaDOGAGA(d)      (c) RaDOGAGA(log(d))

Figure 5: Plot of of $Px(\mathbf{x})$ (x-axis) and $Pz_\psi(\mathbf{z})$ (y-axis). In RaDOGAGA, $Px(\mathbf{x})$ and $Pz_\psi(\mathbf{z})$ are proportional while we can't observe that in DAGMM. Thanks to this trait, we can estimate $Px(\mathbf{x})$ directly from $Pz_\psi(\mathbf{z})$ without ELBO

### 5.1.2 RESULT

Figure 4 displays the distribution of input data source $s$ and latent variable $z$. Even though both methods can capture that $s$ is generated from three mixture-component, there is a difference in PDF. Since the data is generated from GMM, the sample getting closer to the center of clustering, the PDF getting higher. Although, in DAGMM, this tendency looks distorted. This difference is further demonstrated in Fig 5 which shows a plot of $Px(\mathbf{x})$(x-axis) against $Pz_\psi(\mathbf{z})$(y-axis). In our method, $Px(\mathbf{x})$ and $Pz_\psi(\mathbf{z})$ are approximately proportional to each other as in theory while we

cannot observe such proportionality in DAGMM. This difference is quantitatively obvious as well. That is, correlation coefficients between $Px(\boldsymbol{x})$ and $Pz_\psi(\boldsymbol{z})$ are 0.691 (DAGMM), 0.997 (RaDO-GAGA(d)), and 0.999 (RaDOGAGA(log(d))). It also turns out that the correlation coefficient of RaDOGAGA(log(d)) is higher than that of RaDOGAGA(d). Actually, when $Pz_\psi(\boldsymbol{z})$ is sufficiently fitted, $h(d) = log(d)$ makes $Px(\boldsymbol{x})$ and $Pz_\psi(\boldsymbol{z})$ be proportional more rigidly. On the other hand, $h(d) = d$ makes the scale of latent space slightly bent in order to minimize entropy function, allowing relaxed fitting of $Pz_\psi(\boldsymbol{z})$. More detail is described in Appendix D.

## 5.2 ANOMALY DETECTION TASK USING REAL DATA

In this section, we examine whether the clear relationship between $Px(\boldsymbol{x})$ and $Pz(\boldsymbol{z})$ is useful in the anomaly detection task using real data. We use four public datasets[1], KDDCUP99, Thyroid, Arrhythmia, and KDDCUP-Rev. The (instance number, dimension, anomaly ratio(%)) of each dataset is (494021, 121, 20), (3772, 6, 2.5), (452, 274, 15), and (121597, 121, 20) respectively. Detail of datasets is described in Appendix E.

### 5.2.1 EXPERIMENTAL SETUP

We follow the setting in Zong et al. (2018). Randomly extracted 50% of the data is assigned to training and the rest to testing. During training, only normal data is used. During the test, the $R$ for each sample is calculated as the anomaly score, and if the anomaly score is higher than a threshold, it is detected as an anomaly. The threshold is determined by the ratio of anomaly data in each data set. For example, in KDDCup99, data with $R$ in the top 20 % is detected as an anomaly. As metrics, precision, recall, and F1 score are calculated. We run 20 times for each dataset split by 20 different random seeds.

### 5.2.2 BASELINE MODEL

Same as in the previous section, DAGMM is taken as the baseline method. We also compare with the scores reported in previous works in which same experiments were conducted (Zenati et al., 2018; Song & Ou, 2018; Liao et al., 2018).

### 5.2.3 CONFIGURATION

As in Zong et al. (2018), in addition to the output from the encoder, $\frac{\|x-x'\|_2}{\|x\|_2}$ and $\frac{x \cdot x'}{\|x\|_2 \|x'\|_2}$ are concatenated to $\boldsymbol{z}$. It is sent to EN. Note that $\boldsymbol{z}$ is sent to the decoder before concatenation. Other configuration except for hyper parameter is same as in the previous experiment. Hyper parameter for each dataset is described in Appendix E.

### 5.2.4 RESULTS

Table 1 reports the average scores and standard deviations (in brackets). Comparing DAGMM and RaDOGAGA, RaDOGAGA has a better performance regardless of types of $h(\cdot)$. Note that, our model does not increase model complexity at all. Simply introducing the RDO mechanism into the autoencoder has a valid efficacy for anomaly detection. Moreover, our approach achieves state-of-the-art performance compared to other previous works in which same datasets is used. Clear relationship between $Px(\boldsymbol{x})$ and $Pz(\boldsymbol{z})$ by our model is considered to be effective in the task of anomaly detection where the estimating probability distribution is important. In RaDOGAGA, when we compare result of RaDOGAGA(d) and RaDOGAGA(log(d)), either of one is not always superior. As described in section 5.1 and Appendix D, $h(\cdot)$ can be an option depending on fitting flexibility of $Pz(\boldsymbol{z})$.

## 5.3 QUANTIFYING THE IMPACT OF LATENT VARIABLES ON METRICS AND BEHAVIOR AS PCA

In this section, we confirm that (i) the latent variables in our model behaves as PCA components and (ii) the impact of each latent variable on the metrics function can be quantified. When $d\boldsymbol{z}$ in Eq.

---

[1]Dataset can be dowonload from (https://kdd.ics.uci.edu/) and (http://odds.cs.stonybrook.edu)

Table 1: Average and standard deviations(in brackets) of Precision, Recall and F1

| Dataset | Methods | Precision | Recall | F1 |
|---|---|---|---|---|
| KDDCup | ALAD* | 0.9427(0.0018) | 0.9577(0.0018) | 0.9501(0.0018) |
| | INRF* | 0.9452(0.0105) | 0.9600(0.0113) | 0.9525(0.0108) |
| | VAE* | 0.7805 | 0.7903 | 0.7854 |
| | GMVAE* | 0.952 | 0.9141 | 0.9326 |
| | DAGMM* | 0.9297 | 0.9442 | 0.9369 |
| | DAGMM+[†] | 0.9427(0.0052) | 0.9575(0.0053) | 0.9500(0.0052) |
| | RaDOGAGA(d) | 0.9550(0.0037) | 0.9700(0.0038) | 0.9624(0.0038) |
| | RaDOGAGA(log(d)) | **0.9563(0.0042)** | **0.9714(0.0042)** | **0.9638(0.0042)** |
| Thyroid | GMVAE* | **0.7105** | 0.5745 | 0.6353 |
| | VAE* | 0.3395 | 0.3592 | 0.3491 |
| | DAGMM* | 0.4766 | 0.4834 | 0.4782 |
| | DAGMM+[†] | 0.4656(0.0481) | 0.4859(0.0502) | 0.4755(0.0491) |
| | RaDOGAGA(d) | 0.6313(0.0476) | 0.6587(0.0496) | 0.6447(0.0486) |
| | RaDOGAGA(log(d)) | 0.6562(0.0572) | **0.6848(0.0597)** | **0.6702(0.0585)** |
| Arrythmia | ALAD* | 0.5000(0.0208) | 0.5313(0.0221) | 0.5152(0.0214) |
| | VAE* | 0.3328 | 0.3392 | 0.3360 |
| | GMVAE* | 0.4375 | 0.4242 | 0.4308 |
| | DAGMM* | 0.4909 | 0.5078 | 0.4983 |
| | DAGMM+[†] | 0.4985(0.0389) | 0.5136(0.0401) | 0.5060(0.0395) |
| | RaDOGAGA(d) | **0.5353(0.0461)** | **0.5515(0.0475)** | **0.5433(0.0468)** |
| | RaDOGAGA(log(d)) | 0.5294(0.0405) | 0.5455(0.0418) | 0.5373(0.0411) |
| KDDCup-rev | DAGMM* | 0.937 | 0.939 | 0.938 |
| | DAGMM+[†] | 0.9778(0.0018) | 0.9779(0.0017) | 0.9779(0.0018) |
| | RaDOGAGA(d) | 0.9768(0.0033) | 0.9827(0.0012) | 0.9797(0.0015) |
| | RaDOGAGA(log(d)) | **0.9864(0.0009)** | **0.9865(0.0009)** | **0.9865(0.0009)** |

*Score is cited from Zenati et al. (2018)(ALAD), Song & Ou (2018)(INRF), Liao et al. (2018)(VAE, GM-VAE) and Zong et al. (2018) (DAGMM) respectively.

[†]DAGMM+ is our implementation. Note that we also test same configuration as in Zong et al. (2018) and achieve similar score as reported (shown in Appendix E).

(12) is approximated by a small displacement $\boldsymbol{\delta}$, the ratio $D(g(\boldsymbol{z}), g(\boldsymbol{z} + \boldsymbol{\delta}))/\|\boldsymbol{\delta}\|^2$ will be almost constant regardless of $\boldsymbol{z}$ and $\boldsymbol{\delta}$. If the ratio of each component $z_i$ in the latent space is equivalent, the latent space can be regarded as isometric. So, we evaluate this ratio in each component. Let $\boldsymbol{\delta}_i$ be a vector ${}^t(0, \cdots, \epsilon, \cdot, 0)$ where only $i$-th component has a minute value $\epsilon$. Then $D'_i(\boldsymbol{z})$ denotes $\frac{D(g(\boldsymbol{z}), g(\boldsymbol{z}+\boldsymbol{\delta}_i))}{\epsilon^2}$ for $i$-th component. In the trained model, we encode the image $\boldsymbol{x}_l$ and obtain $\boldsymbol{z}_l$. Then, $D'_i(\boldsymbol{z}_l)$ is calculated for each sample. Finally, the average across all samples is measured. This operation is conducted to each dimension of $\boldsymbol{z}$ independently. We also observe the distribution of $\boldsymbol{z}$ and how the image looks different in response to the variation of component $z_i$.

To train model, we use CelebA dataset (Liu et al., 2015), which consists of 202,599 celebrity images. The images are center-cropped so that the image size is 64 x 64.

### 5.3.1 CONFIGURATION

In this experiment, factorized distributions (Ballé et al., 2018) is used to estimate $Pz_\psi(\boldsymbol{z})$. For comparison, we evaluate beta-VAE. Both models are constructed with same depth of CNN and FC layers, with 256-dimensional $\boldsymbol{z}$. Detail of networks and hyper parameter is written in Appendix F. For RaDOGAGA, we set $D(\boldsymbol{x}_1, \boldsymbol{x}_2)$ as $1 - SSIM(\boldsymbol{x}_1, \boldsymbol{x}_2)$ and $h$ as $h(d) = d$. For beta-VAE, reconstruction loss is also $1 - SSIM(\boldsymbol{x}_1, \boldsymbol{x}_2)$. $SSIM(\boldsymbol{x}_1, \boldsymbol{x}_2)$ is defined by Eq. (61).

### 5.3.2 RESULT

Both models are trained so that the SSIM between input and reconstructed image is around 0.93. Figure 6a and 6b show the variance of each component in the latent variables $\boldsymbol{z}$ arranged in descending order. The red line is the cumulative relative ratio of the variance. In Fig. 6b, variance is concentrated in a specific dimension. On the other hand, in Fig. 6a, beta-VAE is trained so that each

latent variable is fitted to a Gaussian distribution with mean 0 and variance 1, there is no significant difference in the variance of each latent variable. Figures 6c and 6d respectively depicts the average of $D_i'(z)$ of each of the top nine dimensions with the largest variance of $z$. In beta-VAE, the $D_i'(z)$ varies drastically depending on the dimension $i$ which shows anisometric latent space, while in RaDOGAGA, it is approximately constant regardless of $i$ which shows isometric latent space. Figure 7 shows decoder outputs when each component $z_i$ is traversed from $-2\sigma$ to $2\sigma$, fixing rest of $z$ as mean. From the top, each row corresponds to $z_0$, $z_1$, $z_2$ ..., and the center column is mean. In Fig. 7b, the image changes visually in any dimension of $z$, while in Fig. 7a, depending on the dimension $i$, there are cases where no significant changes can be seen (such as $z_3$, $z_4$, and so on). This result means that, in RaDOGAGA, the variance of $z$ directly corresponds to the visual impact and the metrics $D(x_1, x_2)$, behaving as PCA. Besides, since $D_i'(z)$ is constant, the latent space shows isometric feature and the variance is regarded as a quantitative importance.

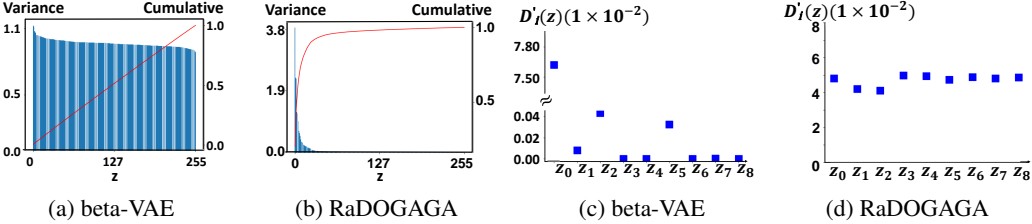

| (a) beta-VAE | (b) RaDOGAGA | (c) beta-VAE | (d) RaDOGAGA |

Figure 6: Variance of $z$ (two on the left) and $D_i'(z)$(two on the right). Compared with beta-VAE, in our model, the variance is concentrated in a few dimensions. Furthermore, the influence each z to the real image is constant. These results demonstrate that the latent variable in our model behaves as PCA components and the importance for the domain data can be quantified.

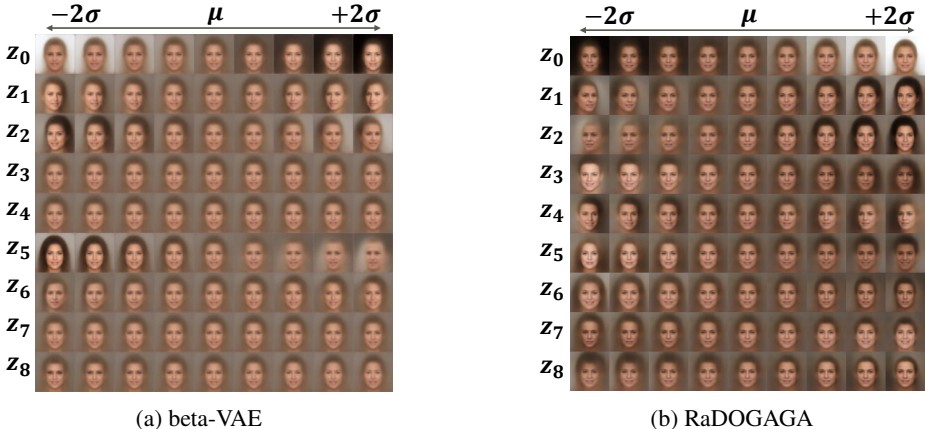

| (a) beta-VAE | (b) RaDOGAGA |

Figure 7: Latent space traversal of z with top-9 variance. In beta-VAE, the variance of z is uncorrelated to difference of image, while they are directly related in our model.

## 6 CONCLUSION

In this paper, we propose RaDOGAGA that learns parametric probability distribution and autoencoder simultaneously based on rate-distortion optimization. It was proven that the probability distribution of the latent variables obtained by the proposed method is proportional to the probability distribution of the input data theoretically and experimentally. This property is validated in anomaly detection achieving state-of-the-art performance. Moreover, our model has the trait as PCA which likely promotes interpretation of latent variables. For the future work, we will conduct experiments with different types of metrics functions that derived from semantical task, such as in categorical classification. Meanwhile, as mentioned in Tschannen et al. (2019), considering the usefulness of latent variables in downstream task is another research direction to explore.

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

# A HOW JACOBI MATRIX BECOME A CONSTANTLY SCALED ORTHONORMAL BASIS

In this appendix section, we prove that all of column vectors in decoder's Jacobi Matrix have the same norm and are orthogonal to each other. In other words, each column of Jacobi matrix is a constantly scaled orthonormal basis.

Here we assume that data space sample $x$ has $M$ dimension, that is $x \in \mathbf{R}^M$, then encoded to $N$ dimensional latent space sample $y \in \mathbf{R}^N$. We also assume that an fixed encoder function $y = f_{init,\theta}(x)$ and a fixed decoder $\hat{x} = g_{init,\phi}(y)$ are given such that $h(D(x, \hat{x}))$ becomes minimal.

$$y = f_{init,\theta}(x) \tag{22}$$
$$\hat{x} = g_{init,\phi}(y) \tag{23}$$
$$s.t \ \ h(D(x, \hat{x})) \implies \text{minimal}$$

We further assume that fixed parametric PDF of latent variable $y$ is also given.

$$P y_{init,\psi}(y) \tag{24}$$

Here, it is noted that this function is not needed to be optimal in a sense of $D_{KL}\left(Py\left(y\right) \| Py_{init,\psi}\left(y\right)\right)$ where $Py\left(y\right)$ is an actual PDF of $y$.

Under these conditions, we introduce new latent variable $z$, and $y$ is transformed from $z$ using a following scaling function $a(z) : \mathbf{R}^N \rightarrow \mathbf{R}^N$.

$$y = a(z) = (a_1(z), a_2(z), \cdots, a_N(z)) \tag{25}$$

Here, **our goal** is to prove Eq. (10) by examining the condition of this scaling function which minimize the average of Eq. (5) with regard to $\epsilon \sim P(\epsilon)$.

Because of the assumption of minimal $D(x, \hat{x})$, we ignore the second term of Eq. (5). Next, the PDF of $z$ can be derived using a Jacobian of $a(z)$.

$$Pz_{\psi,a}(z) = \left| \frac{\mathrm{d}a(z)}{\mathrm{d}z} \right| \cdot Py_{init,\psi}(a(z)) \tag{26}$$

By applying these conditions and equations to Eq. (5), the cost of scaling function $a(z)$ to minimize the average is expressed as follows.

$$L_a = -\log\left(\left| \frac{\mathrm{d}a(z)}{\mathrm{d}z} \right| \cdot Py_{init,\psi}(a(z))\right) + \lambda_2 \cdot D(g_{init,\phi}(a(z + \epsilon)),\ g_{init,\phi}(a(z))) \tag{27}$$

Next, the latent variable is fixed as $y_0$, and $z_0$ is defined in the next equation.

$$z_0 = a^{-1}(y_0) \tag{28}$$

Then the average of Eq. (27) with regard to $\epsilon \sim P(\epsilon)$ is expressed as follows.

$$E_{\epsilon \sim P(\epsilon)}\left[L_a|_{z=z_0}\right] = -\log\left(\left| \frac{\mathrm{d}a(z)}{\mathrm{d}z} \right|_{z=z_0} \cdot Py_{init,\psi}(a(z_0))\right)$$
$$+ \lambda_2 \cdot E_{\epsilon \sim P(\epsilon)}\left[D(g_{init,\phi}(a(z_0 + \epsilon)),\ g_{init,\phi}(a(z_0)))\right] \tag{29}$$

Then the condition of $a(z_0)$ in the neighborhood of $y_0$ is examined when Eq. (29) is minimized. Here, it is noted that the first term of the right side doesn't depend on $\epsilon$.

Before examining Eq. (29), some preparation of equations are needed. At first, Jacobi matrix of $\boldsymbol{a}(\boldsymbol{z})$ at $\boldsymbol{z} = \boldsymbol{z}_0$ is defined as $\boldsymbol{B}$ using notations of partial differentials $b_{ij} = \left.\frac{\partial a_i}{\partial z_j}\right|_{\boldsymbol{z}=\boldsymbol{z}_0}$.

$$
\boldsymbol{B} = \left.\frac{\partial \boldsymbol{y}}{\partial \boldsymbol{z}}\right|_{\boldsymbol{z}=\boldsymbol{z}_0} = \left.\frac{\mathrm{d}\boldsymbol{a}(\boldsymbol{z})}{\mathrm{d}\boldsymbol{z}}\right|_{\boldsymbol{z}=\boldsymbol{z}_0} = \begin{pmatrix} b_{11} & b_{12} & \dots & b_{1N} \\ b_{21} & b_{22} & \dots & b_{2N} \\ \vdots & \vdots & \ddots & \vdots \\ b_{N1} & b_{N2} & \dots & b_{NN} \end{pmatrix}
$$
$$
where \; b_{ij} = \left.\frac{\partial a_i}{\partial z_j}\right|_{\boldsymbol{z}=\boldsymbol{z}_0} \tag{30}
$$

The vector $\boldsymbol{b}_i$ is also defined as follows.

$$
\boldsymbol{b}_i = \left.\frac{\partial \boldsymbol{y}}{\partial z_j}\right|_{\boldsymbol{z}=\boldsymbol{z}_0} = \begin{pmatrix} b_{1i} \\ b_{2i} \\ \vdots \\ b_{Ni} \end{pmatrix} \tag{31}
$$

It is clear by definition that $\boldsymbol{b}_i$ and $\boldsymbol{B}$ have the following relation.

$$
\boldsymbol{B} = (\boldsymbol{b}_1, \boldsymbol{b}_2, \cdots, \boldsymbol{b}_N) \tag{32}
$$

$\Delta_{ij}$ is defined as a cofactor of matrix $\boldsymbol{B}$ with regard to the element $b_{ij}$, and $\tilde{\boldsymbol{b}}_i$ is also defined by the following equation.

$$
\tilde{\boldsymbol{b}}_i = \begin{pmatrix} \Delta_{1i} \\ \Delta_{2i} \\ \vdots \\ \Delta_{Ni} \end{pmatrix} \tag{33}
$$

The following equations hold by the definition of cofactor.

$$
{}^t\boldsymbol{b}_i\,\tilde{\boldsymbol{b}}_i \;=\; \sum_{k=1}^{N} b_{ki} \cdot \Delta_{ki} = |\boldsymbol{B}| \tag{34}
$$

$$
\frac{\mathrm{d}|\boldsymbol{B}|}{\mathrm{d}b_{ij}} \;=\; \Delta_{ij}, \quad \frac{\mathrm{d}|\boldsymbol{B}|}{\mathrm{d}\boldsymbol{b}_i} = \tilde{\boldsymbol{b}}_i \tag{35}
$$

In case of $i \neq j$, inner product of $\boldsymbol{b}_i$ and $\tilde{\boldsymbol{b}}_j$ becomes zero because this value is a determinant of a singular matrix with two equivalent column vectors $\boldsymbol{b}_i$.

$$
{}^t\boldsymbol{b}_i\,\tilde{\boldsymbol{b}}_j = \sum_{k=1}^{N} b_{ki} \cdot \Delta_{kj} = |(\boldsymbol{b}_1, \cdots, \boldsymbol{b}_i, \cdots, \boldsymbol{b}_i, \cdots)| = 0 \tag{36}
$$

$\boldsymbol{G}'_{init}$ is defined as a $M \times N$ Jacobi matrix of $g_{init,\phi}(\boldsymbol{y})$ at $\boldsymbol{y} = \boldsymbol{y}_0$ as follows.

$$
\boldsymbol{G}'_{init} = \left.\frac{\mathrm{d}g_{init,\phi}(\boldsymbol{y})}{\mathrm{d}\boldsymbol{y}}\right|_{\boldsymbol{y}=\boldsymbol{y}_0} \tag{37}
$$

Using these equations, we proceed to expand of Eq. (29). The first term of (29) in the right side can be expressed as follows, where the second term Eq. (38) in the right side is constant by definition.

$$
-\log\left(\left|\left.\frac{\mathrm{d}\boldsymbol{a}(\boldsymbol{z})}{\mathrm{d}\boldsymbol{z}}\right|_{\boldsymbol{z}=\boldsymbol{z}_0}\right| \cdot P\boldsymbol{y}_{init,\psi}(\boldsymbol{a}(\boldsymbol{z}_0))\right) = -\log(|\boldsymbol{B}|) - \log(P\boldsymbol{y}_{init,\psi}(\boldsymbol{y}_0)) \tag{38}
$$

Next step is an expansion of the second term in Eq. (29). First, the following approximation holds.

$$
\begin{aligned}
g_{init,\phi}(\boldsymbol{a}(\boldsymbol{z}_0 + \boldsymbol{\epsilon})) - g_{init,\phi}(\boldsymbol{a}(\boldsymbol{z}_0)) \quad &\sim \quad \left( \left. \frac{\mathrm{d}g_{init,\phi}(\boldsymbol{y})}{\mathrm{d}\boldsymbol{y}} \right|_{\boldsymbol{y}=\boldsymbol{y}_0} \right) \left( \left. \frac{\mathrm{d}\boldsymbol{a}(\boldsymbol{z})}{\mathrm{d}\boldsymbol{z}} \right|_{\boldsymbol{z}=\boldsymbol{z}_0} \right) \boldsymbol{\epsilon} \\
&= \quad \boldsymbol{G}'_{init} \, \boldsymbol{B} \, \boldsymbol{\epsilon} \\
&= \quad \boldsymbol{G}'_{init} \, (\epsilon_1 \cdot \boldsymbol{b}_1 + \epsilon_2 \cdot \boldsymbol{b}_2 \cdots + \epsilon_N \cdot \boldsymbol{b}_N) \\
&= \quad \sum_{i=1}^{N} \epsilon_i \cdot \boldsymbol{G}'_{init} \boldsymbol{b}_i
\end{aligned}
\tag{39}
$$

Then, the second term of Eq. (29) can be transformed to the next equation by using Eqs. (39), (3), and (6).

$$
\begin{aligned}
E_{\boldsymbol{\epsilon}\sim P(\boldsymbol{\epsilon})} \left[ D(g_{init,\phi}(\boldsymbol{a}(\boldsymbol{z}_0 + \boldsymbol{\epsilon})), g_{init,\phi}(\boldsymbol{a}(\boldsymbol{z}_0))) \right] \quad &\sim \quad E_{\boldsymbol{\epsilon}\sim P(\boldsymbol{\epsilon})} \left[ \left\| \boldsymbol{L}(\boldsymbol{x}) \sum_{i=1}^{N} \epsilon_i \cdot \boldsymbol{G}'_{init} \, \boldsymbol{b}_i \right\|^2 \right] \\
&= \quad \sum_{i=1}^{N} E[\epsilon_i^2] \cdot \| \boldsymbol{L}(\boldsymbol{x}) \, \boldsymbol{G}'_{init} \, \boldsymbol{b}_i \|^2 \\
&\quad + 2 \cdot \sum_{i=1}^{N} \sum_{j=i+1}^{N} E[\epsilon_i \cdot \epsilon_j] \cdot \\
&\qquad {}^t(\boldsymbol{L}(\boldsymbol{x}) \, \boldsymbol{G}'_{init} \, \boldsymbol{b}_i)(\boldsymbol{L}(\boldsymbol{x}) \, \boldsymbol{G}'_{init} \, \boldsymbol{b}_j) \\
&= \quad \sigma^2 \cdot \left( \sum_{i=1}^{N} \| \boldsymbol{L}(\boldsymbol{x}) \, \boldsymbol{G}'_{init} \, \boldsymbol{b}_i \|^2 \right)
\end{aligned}
\tag{40}
$$

As a result, Eq. (29) can be rewritten as Eq. (41).

$$
\begin{aligned}
E_{\boldsymbol{\epsilon}\sim P(\boldsymbol{\epsilon})} \left[ L_a |_{\boldsymbol{z}=\boldsymbol{z}_0} \right] \quad &\sim \quad -\log(|\boldsymbol{B}|) - \log(P\boldsymbol{y}_{init,\psi}(\boldsymbol{y}_0)) \\
&\quad + \lambda_2 \cdot \sigma^2 \cdot \left( \sum_{i=1}^{N} \| \boldsymbol{L}(\boldsymbol{x}) \, \boldsymbol{G}'_{init} \, \boldsymbol{b}_i \|^2 \right)
\end{aligned}
\tag{41}
$$

By examining the minimization process of Eq. (41), the conditions of optimal scaling function $\boldsymbol{y} = \boldsymbol{a}(\boldsymbol{z})$ in the neighborhood of $\boldsymbol{y}_0$ is clarified. Here, the condition of Jacobi matrix $\boldsymbol{B}$ is examined instead of $\boldsymbol{a}(\boldsymbol{z})$. Eq. (41) is differentiated by vector $\boldsymbol{b}_i$, and the result is set to be zero. Then the following equation Eq.(42) is derived.

$$
2\lambda_2 \cdot \sigma^2 \cdot \left( {}^t(\boldsymbol{L}(\boldsymbol{x}) \, \boldsymbol{G}'_{init})(\boldsymbol{L}(\boldsymbol{x}) \, \boldsymbol{G}'_{init}) \right) \boldsymbol{b}_i = \frac{1}{|\boldsymbol{B}|} \cdot \tilde{\boldsymbol{b}}_i
\tag{42}
$$

Afterwards, Eq. (42) is multiplied by ${}^t\boldsymbol{b}_j$ from the left, and divided by $2\lambda_2 \cdot \sigma^2$. As a result, Eq. (43) is derived by using Eqs. (34) and (36).

$$
{}^t(\boldsymbol{L}(\boldsymbol{x}) \, \boldsymbol{G}'_{init} \, \boldsymbol{b}_j) \, (\boldsymbol{L}(\boldsymbol{x}) \, \boldsymbol{G}'_{init} \, \boldsymbol{b}_i) = \frac{1}{2\lambda_2 \cdot \sigma^2} \cdot \delta_{ij}
\tag{43}
$$

Here, we define the following function $g_{ortho,\psi}(\boldsymbol{z})$ which is a composite function of $g_{init,\psi}()$ and $\boldsymbol{a}()$.

$$
\hat{\boldsymbol{x}} = g_{ortho,\psi}(\boldsymbol{z}) = g_{init,\psi}(\boldsymbol{a}(\boldsymbol{z}))
\tag{44}
$$

Then the next equation holds by definition.

$$
\left. \frac{\partial g_{ortho,\psi}(\boldsymbol{z})}{\partial z_i} \right|_{\boldsymbol{z}=\boldsymbol{z}_0} = \left. \frac{\partial \hat{\boldsymbol{x}}}{\partial z_i} \right|_{\boldsymbol{z}=\boldsymbol{z}_0} = \boldsymbol{G}'_{init} \, \boldsymbol{b}_i
\tag{45}
$$

It is noted that this equation holds at any value of $\boldsymbol{y}_0$ or $\boldsymbol{z}_0$. As a result, the following equation, that is Eq. (10), can be derived.

$$^t\left(\boldsymbol{L}(\boldsymbol{x})\frac{\partial\hat{\boldsymbol{x}}}{\partial z_i}\right)\left(\boldsymbol{L}(\boldsymbol{x})\frac{\partial\hat{\boldsymbol{x}}}{\partial z_j}\right) = \frac{1}{2\lambda_2\sigma^2}\cdot\delta_{ij} \tag{46}$$

If encoder and decoder are trained well and $\boldsymbol{x}\simeq\hat{\boldsymbol{x}}$ holds, we can introduce new rescaled data space $\boldsymbol{x}_D$ determined by metrics function such as $\mathrm{d}\boldsymbol{x}_D = \boldsymbol{L}(\boldsymbol{x})\cdot\mathrm{d}\boldsymbol{x}$, and the next equation holds.

$$^t\left(\frac{\partial\boldsymbol{x}_D}{\partial z_i}\right)\left(\frac{\partial\boldsymbol{x}_D}{\partial z_j}\right) = \frac{1}{2\lambda_2\sigma^2}\cdot\delta_{ij} \tag{47}$$

In conclusion, all column vectors of Jacobi matrix between $\boldsymbol{z}$ and $\boldsymbol{x}_D$ has the same L2 norm $1/\sqrt{2\lambda_2}\sigma$ and all pairs of column vectors are orthogonal. In other words, when column vectors of Jacobi matrix are multiplied by the constant $\sqrt{2\lambda_2}\sigma$, the resulting vectors are orthonormal.

# B  EXPLANATION OF "CONTINUOUS PCA" FEATURE

In this section, we explain RaDOGAGA has a continuous PCA feature when factorized probability density model as below is used.

$$P\boldsymbol{z}_\psi(\boldsymbol{z}) = \prod_{i=1}^{N} Pz_{i\psi}(z_i) \tag{48}$$

Here, our definition of "continuous PCA" feature means the following. 1) Mutual information between latent variables are minimum and likely to be uncorrelated to each other. 2) Energy of latent space is concentrated to several principal components, and the importance of each component can be determined.

Next we explain the reason why these feature is acquired. As explained in appendix A, all column vectors of Jacobi matrix of decoder from latent space to data space have the same norm and all combinations of pairwise vectors are orthogonal. In other words, when constant value is multiplied, the resulting vectors are orthonormal. Because encoder is a inverse function of decoder ideally, each row vector of encoder's Jacobi matrix should be the same as column vector of decoder under the ideal condition. Here, $f_{ortho,\theta}(\boldsymbol{x})$ and $g_{ortho,\phi}(\boldsymbol{z}_\theta)$ are defined as encoder and decoder with these feature. Because the latent variables depend on encoder parameter $\theta$, latent variable is described as $\boldsymbol{z}_\theta = f_{ortho,\theta}(\boldsymbol{x})$, and its PDF is defined as $P\boldsymbol{z}_\theta(\boldsymbol{z}_\theta)$. PDFs of latent space and data space have the following relation where $J$ is a Jacobian or pseudo-Jacobian between two spaces with constant value as explained in appendix A.

$$P\boldsymbol{z}_\theta(\boldsymbol{z}_\theta) \simeq J\cdot P\boldsymbol{x}_D(\boldsymbol{x}_D) \propto P\boldsymbol{x}_D(\boldsymbol{x}_D) \tag{49}$$

As described before, $P\boldsymbol{z}_\psi(\boldsymbol{z})$ is a parametric PDF of the latent space to be optimized with parameter $\psi$.

By applying the result of Eqs. (41) and (43), Eq. (5) can be transformed as Eq. (50) where $\hat{\boldsymbol{x}} = g_{ortho,\phi}(f_{ortho,\theta}(\boldsymbol{x}))$.

$$L_{ortho} = -\log\left(P\boldsymbol{z}_\psi(\boldsymbol{z}_\theta)\right) + \lambda_1\cdot h\left(D(\boldsymbol{x},\hat{\boldsymbol{x}})\right) + N/2$$
$$s.t. \quad ^t\left(\frac{\partial g_{ortho,\phi}(\boldsymbol{z}_\theta)}{\partial z_{\theta_i}}\right)\left(\frac{\partial g_{ortho,\phi}(\boldsymbol{z}_\theta)}{\partial z_{\theta_j}}\right) = \frac{1}{2\lambda\sigma^2}\cdot\delta_{ij} \tag{50}$$

Here, the third term of the right side is constant, this term can be removed from the cost function as follows.

$$L'_{ortho} = -\log\left(P\boldsymbol{z}_\psi(\boldsymbol{z}_\theta)\right) + \lambda_1\cdot h\left(D(\boldsymbol{x},\hat{\boldsymbol{x}})\right) \tag{51}$$

Then the parameters of network and probability can be obtained by the next.

$$\theta,\phi,\psi = \underset{\theta,\phi,\psi}{\arg\min}(E_{\boldsymbol{x}\sim P\boldsymbol{x}(\boldsymbol{x})}[L'_{ortho}]) \tag{52}$$

$E_{\boldsymbol{x}\sim P\boldsymbol{x}(\boldsymbol{x})}[L'_{ortho}]$ in Eq. （52）can be transformed as the next.

$$
\begin{aligned}
E_{\boldsymbol{x}\sim P\boldsymbol{x}(\boldsymbol{x})}[L'_{ortho}] &= \int P\boldsymbol{x}(\boldsymbol{x})\cdot\left(-\log\left(P\boldsymbol{z}_\psi(\boldsymbol{z}_\theta)\right)+\lambda_1\cdot h\left(D(\boldsymbol{x},\hat{\boldsymbol{x}})\right)\right)\mathrm{d}\boldsymbol{x} \\
&= \int\left(P\boldsymbol{z}_\theta(\boldsymbol{z}_\theta)\cdot\left|\frac{\mathrm{d}\boldsymbol{x}}{\mathrm{d}\boldsymbol{z}_\theta}\right|^{-1}\right)\cdot\left(-\log\left(P\boldsymbol{z}_\psi(\boldsymbol{z}_\theta)\right)\right)\cdot\left|\frac{\mathrm{d}\boldsymbol{x}}{\mathrm{d}\boldsymbol{z}_\theta}\right|\mathrm{d}z_\theta \\
&\quad +\lambda_1\cdot\int P\boldsymbol{x}_D(\boldsymbol{x}_D)\cdot|\boldsymbol{L}(\boldsymbol{x})|^{-1}\cdot h\left(D(\boldsymbol{x},\hat{\boldsymbol{x}})\right)\cdot|\boldsymbol{L}(\boldsymbol{x})|\mathrm{d}\boldsymbol{x}_D \\
&= \int P\boldsymbol{z}_\theta(\boldsymbol{z}_\theta)\cdot\left(-\log\left(P\boldsymbol{z}_\psi(\boldsymbol{z}_\theta)\right)\right)\mathrm{d}z_\theta \\
&\quad +\lambda_1\cdot\int P\boldsymbol{x}_D(\boldsymbol{x}_D)\cdot h\left(D(\boldsymbol{x},\hat{\boldsymbol{x}})\right)\mathrm{d}\boldsymbol{x}_D \qquad (53)
\end{aligned}
$$

At first, the first term of the third formula in Eq.(53) is examined. Let $\mathrm{d}\boldsymbol{z}_{\theta/i}$ be a differential of $(N-1)$ dimensional latent variables where i-th axis $z_{\theta i}$ is removed from the latent variable $\boldsymbol{z}_\theta$. Then a marginal distribution of $z_{\theta i}$ can be derived from the next equation.

$$
Pz_{\theta i}(z_{\theta i}) = \int P\boldsymbol{z}_\theta(\boldsymbol{z}_\theta)\mathrm{d}\boldsymbol{z}_{\theta/i} \qquad (54)
$$

By using Eqs.(48) and (54), the first term of the third formula in Eq. (53) can be expanded as follows.

$$
\begin{aligned}
\int P\boldsymbol{z}_\theta(\boldsymbol{z}_\theta)\cdot\left(-\log\left(P\boldsymbol{z}_\psi(\boldsymbol{z}_\theta)\right)\right)\mathrm{d}\boldsymbol{z}_\theta &= \int P\boldsymbol{z}_\theta(\boldsymbol{z}_\theta)\cdot\left(-\log\left(\frac{\prod_{i=1}^N Pz_{i\psi}(z_{\theta i})}{\prod_{i=1}^N Pz_{\theta i}(z_{\theta i})}\right)\right)\mathrm{d}\boldsymbol{z}_\theta \\
&\quad +\int P\boldsymbol{z}_\theta(\boldsymbol{z}_\theta)\cdot\left(-\log\left(\prod_{i=1}^N Pz_{\theta i}(z_{\theta i})\right)\right)\mathrm{d}\boldsymbol{z}_\theta \\
&= \sum_{i=1}^N\int\left(\int P\boldsymbol{z}_\theta(\boldsymbol{z}_\theta)\mathrm{d}\boldsymbol{z}_{\theta/i}\right)\cdot\left(-\log\left(\frac{Pz_{i\psi}(z_{\theta i})}{Pz_{\theta i}(z_{\theta i})}\right)\right)\mathrm{d}z_{\theta i} \\
&\quad +\sum_{i=1}^N\int\left(\int P\boldsymbol{z}_\theta(\boldsymbol{z}_\theta)\mathrm{d}\boldsymbol{z}_{\theta/i}\right)\cdot\left(-\log\left(Pz_{\theta i}(z_{\theta i})\right)\right)\mathrm{d}z_{\theta i} \\
&= \sum_{i=1}^N D_{KL}(Pz_{\theta i}(z_{\theta i})\|Pz_{i\psi}(z_{\theta i})) + \sum_{i=1}^N H(z_{\theta i}) \qquad (55)
\end{aligned}
$$

$H(X)$ is an entropy of variable $X$. The first term of the third formula is KL-divergence between marginal probability $Pz_{\theta i}(z_{\theta i})$ and factorized parametric probability $Pz_{i\psi}(z_{\theta i})$. The second term of the third formula can be further transformed using mutual information between latent variables $I(\boldsymbol{z}_\theta)$ and equation (49).

$$
\sum_{i=1}^N H(z_{\theta i}) = H(\boldsymbol{z}_\theta) + I(\boldsymbol{z}_\theta) \simeq H(\boldsymbol{x}_D) - \log(J) + I(\boldsymbol{z}_\theta) \qquad (56)
$$

The first term of the third formula is the entropy of input data with constant value. The second is also constant. As a result, in order to minimize (55), mutual information $I(\boldsymbol{z}_\theta)$ must be minimized.

At second, the second term of the third formula in Eq. (53) is examined. $\boldsymbol{x}_D$ and $\hat{\boldsymbol{x}}_D$ denote mapped values to rescaled data space defined by the metrics function $D(\boldsymbol{x}_1,\boldsymbol{x}_2)$ as Eq.(9). Because $\boldsymbol{x}_D$ and $\hat{\boldsymbol{x}_D}$ are close, the following equations holds.

$$
\begin{aligned}
\hat{\boldsymbol{x}}_D &\sim \boldsymbol{x}_D + \boldsymbol{L}(\boldsymbol{x})\left(\hat{\boldsymbol{x}} - \boldsymbol{x}\right) & (57) \\
D(\boldsymbol{x},\hat{\boldsymbol{x}}) &\sim |\boldsymbol{L}(\boldsymbol{x})\left(\hat{\boldsymbol{x}} - \boldsymbol{x}\right)|^2 \sim |\boldsymbol{x}_D - \hat{\boldsymbol{x}}_D|^2 & (58)
\end{aligned}
$$

By using these expansions, Eq.(53) can be expressed as follows.

$$
\begin{aligned}
E_{\boldsymbol{x} \sim P\boldsymbol{x}(\boldsymbol{x})}[L'_{ortho}] \quad \simeq \quad & \sum_{i=1}^{N} D_{KL}(Pz_{\theta i}(z_{\theta i}) \| Pz_{i\psi}(z_{\theta i})) \\
& + I(\boldsymbol{z}_\theta) + E_{\boldsymbol{x}_D}\left[\, |\boldsymbol{x}_D - \hat{\boldsymbol{x}}_D|^2 \,\right] + \text{Const.}
\end{aligned}
\tag{59}
$$

Here, the rescaled real space $\boldsymbol{x}_D$ is divided into a plurality of small subspace partitionings $\Omega\boldsymbol{x}_{D1}$, $\Omega\boldsymbol{x}_{D2}$, $\cdots$. Let $\Omega\boldsymbol{z}_1$, $\Omega\boldsymbol{z}_2$, $\cdots$ be corresponding subspace partitionings in latent space. as the division space of the latent space $\boldsymbol{z} \in \boldsymbol{R}^N$ corresponding to $\Omega\boldsymbol{x}_D$. Then Eq. (59) can be rewritten as follows.

$$
\begin{aligned}
E_{\boldsymbol{x} \sim P\boldsymbol{x}(\boldsymbol{x})}[L'_{ortho}] \quad \simeq \quad & \sum_{i=1}^{N} D_{KL}(Pz_{\theta i}(z_{\theta i}) \| Pz_{i\psi}(z_{\theta i})) \\
& + \sum_k \left( I(\boldsymbol{z}_\theta \in \Omega\boldsymbol{z}_{\theta k}) + E_{\boldsymbol{x}_D \in \Omega\boldsymbol{x}_{Dk}}\left[\, |\boldsymbol{x}_D - \hat{\boldsymbol{x}}_D|^2 \,\right] \right) + \text{Const.}
\end{aligned}
\tag{60}
$$

For each subspace partitioning, the transformation from $\Omega\boldsymbol{x}_{Dk}$ to $\Omega\boldsymbol{z}_{\theta k}$ can be regarded as constantly scaled orthonormal transformation where orthonormal basis is Jacobi matrix with scale factor $J^{-1}$.

According to Karhunen-Loève Theory (Rao & Yip (2000)), the orthonormal basis which minimize both mutual information and reconstruction error leads to be Karhunen-Loève transform(KLT). It is noted that the basis of KLT is equivalent to PCA orthonormal basis.

As a result, when Eq. (60) is minimized, Jacobi matrix from $\Omega\boldsymbol{x}_{Dk}$ to $\Omega\boldsymbol{z}_{\theta k}$ for each subspace partitioning should be KLT/PCA. Accordingly, the same feature as PCA will be realized such as the determination of principal components etc.

From these consideration, we conclude that RaDOGAGA has a "continuous PCA" feature.

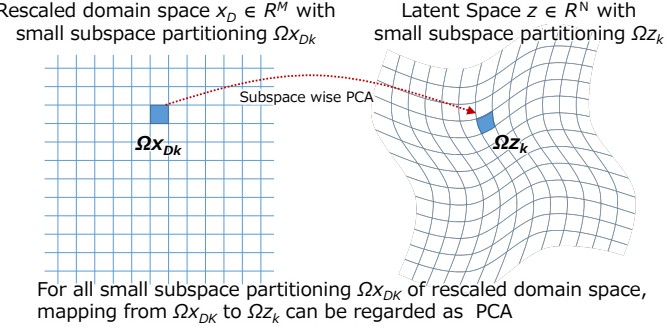

Figure 8: Continuous KLT(PCA) Mapping from input domain to latent space

## C    EXPANSION OF SSIM TO A QUADRATIC FORM

Structural similarity (SSIM) (Wang et al., 2001) is widely used for picture quality metric which is close to human subjective quality. In this appendix, we show $(1 - SSIM)$ can be approximated to a quadratic form such as Eq.(6).

Eq. (61) is a SSIM value for a $N \times N$ window between picture $\boldsymbol{x}$ and $\boldsymbol{y}$. In order to calculate SSIM index for a picture, this window is shifted in a whole picture and all of SSIM values are averaged.

$$
SSIM_{N \times N}(\boldsymbol{x}, \boldsymbol{y}) = \frac{2\mu_x \mu_y}{\mu_x{}^2 + \mu_y{}^2} \cdot \frac{2\sigma_{xy}}{\sigma_x{}^2 + \sigma_y{}^2}
\tag{61}
$$

If $(1 - SSIM_{N \times N}(\boldsymbol{x}, \boldsymbol{y}))$ is expressed in quadratic form, the average for a picture $(1 - SSIM_{picture})$ can be also expressed in quadratic form.

Let $\delta \boldsymbol{x}$ be a minute displacement of $\boldsymbol{x}$. Then SSIM between $\boldsymbol{x}$ and $\boldsymbol{x} + \delta \boldsymbol{x}$ can be approximated as follows

$$SSIM_{N \times N}(\boldsymbol{x}, \boldsymbol{x} + \delta \boldsymbol{x}) = 1 - \frac{\mu_{\delta \boldsymbol{x}}{}^2}{2\mu_x{}^2} - \frac{\sigma_{\delta \boldsymbol{x}}{}^2}{2\sigma_x{}^2} + O\left((|\delta \boldsymbol{x}|/|\boldsymbol{x}|)^3\right) \tag{62}$$

Then $\mu_{\delta \boldsymbol{x}}{}^2$ and $\sigma_{\delta \boldsymbol{x}}{}^2$ can be expressed as follows.

$$\mu_{\delta \boldsymbol{x}}{}^2 = {}^t\delta \boldsymbol{x} \cdot \boldsymbol{M} \cdot \delta \boldsymbol{x}$$

$$where \; \boldsymbol{M} = \frac{1}{N^2} \cdot \begin{pmatrix} 1 & 1 & \dots & 1 \\ 1 & 1 & \dots & 1 \\ \vdots & \vdots & \ddots & \vdots \\ 1 & 1 & \dots & 1 \end{pmatrix} \tag{63}$$

$$\sigma_{\delta \boldsymbol{x}}{}^2 = {}^t\delta \boldsymbol{x} \cdot \boldsymbol{V} \cdot \delta \boldsymbol{x}$$

$$where \; \boldsymbol{V} = \frac{1}{N^2} \cdot \begin{pmatrix} N-1 & -1 & \dots & -1 \\ -1 & N-1 & \dots & -1 \\ \vdots & \vdots & \ddots & \vdots \\ -1 & -1 & \dots & N-1 \end{pmatrix} \tag{64}$$

It is noted that matrix $\boldsymbol{M}$ is positive definite and matrix $\boldsymbol{V}$ is positive semidefinite. As a result, $(1 - SSIM_{N \times N}(\boldsymbol{x}, \boldsymbol{y}))$ can be expressed in the following quadratic form with positive definite matrix.

$$1 - SSIM_{N \times N}(\boldsymbol{x}, \boldsymbol{x} + \delta \boldsymbol{x}) \simeq {}^t\delta \boldsymbol{x} \cdot \left( \frac{1}{2\mu_x{}^2} \cdot \boldsymbol{M} + \frac{1}{2\sigma_x{}^2} \cdot \boldsymbol{V} \right) \cdot \delta \boldsymbol{x} \tag{65}$$

## D    EFFECT OF $h(x)$

In this appendix section, the effects of two kinds of cost scaling function $h(d) = d$ and $h(d) = \log(d)$ are discussed. We evaluated the behaviors of encoder and decoder in a one dimensional model using simple parametric linear encoder and decoder.

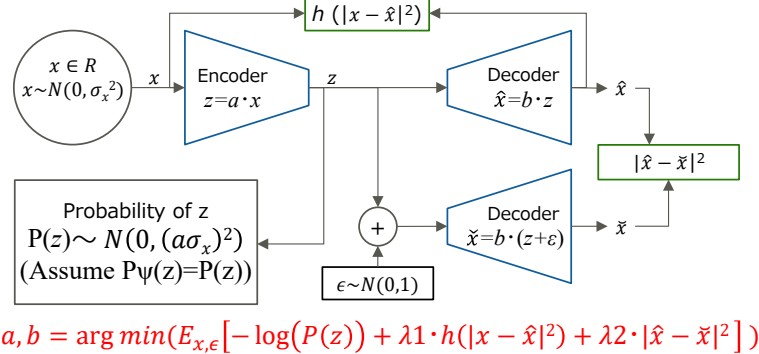

Figure 9: Simple encoder/decoder model to evaluate $h(d)$

Lex $x$ be a one dimensional data with the normal distribution.

$$\begin{aligned} x &\in R \\ x &\sim N(0, \sigma_x{}^2) \end{aligned}$$

Lex $z$ be a one dimensional latent variable. Following two linear encoder and decoder are provided with parameter $a$ and $b$.

$$
\begin{aligned}
z &= a \cdot x \\
\hat{x} &= b \cdot z
\end{aligned}
$$

Here, square error is used as metrics function $D(x, y)$. The distribution of noise $\epsilon$ added to latent variable z is set to $N(0, 1)$. Then $\breve{x}$ is derived by decoding $z + \epsilon$.

$$
\begin{aligned}
D(x, y) &= |x - y|^2 \\
\epsilon &\sim N(0, 1) \\
\breve{x} &= b \cdot (z + \epsilon)
\end{aligned}
$$

For simplicity, we assume parametric PDF $P\psi(z)$ is equal to the real PDF $P(z)$. Because the distribution of latent variable $z$ follows $N(0, (a\sigma_x)^2)$, the entropy of $z$ can be expressed as follows.

$$
\begin{aligned}
P(z) &\sim N(0, (a\sigma_x)^2) \\
H(z) &= \int -P(z) \cdot \log(P(z)) \mathrm{d}z \\
&= \log(a) + \log(\sigma_x \sqrt{2\pi e})
\end{aligned}
$$

Using these notations, Eqs. (5) and (8) can be expressed as follows.

$$
\begin{aligned}
Loss &= E_{x \sim N(0, \sigma_x^2), \ \epsilon \sim N(0, 1)} \left[ -\log P(z) + \lambda_1 \cdot h(|x - \hat{x}|^2) + \lambda_2 \cdot |\hat{x}, \breve{x}|^2 \right] \\
&= \log(a) + \log(\sigma_x \sqrt{2\pi e}) + \lambda_1 \cdot E_{x \sim N(0, \sigma_x^2)} \left[ h(|x - \hat{x}|^2) \right] + \lambda_2 \cdot b^2 \quad (66)
\end{aligned}
$$

At first, the case of $h(d) = d$ is examined. By applying $h(d) = d$, Eq. (66) can be expanded as follows.

$$
Loss = \log(a) + \log(\sigma_x \sqrt{2\pi e}) + \lambda_1 \cdot (a \cdot b - 1)^2 \cdot \sigma_x{}^2 + \lambda_2 \cdot b^2 \quad (67)
$$

By solving $\frac{\partial Loss}{\partial a} = 0$ and $\frac{\partial Loss}{\partial b} = 0$, $a$ and $b$ are derived as follows.

$$
\begin{aligned}
a \cdot b &= \frac{\lambda_1 \sigma_x{}^2 + \sqrt{\lambda_1{}^2 \sigma_x{}^4 - 2\lambda_1 \sigma_x{}^2}}{2\lambda_1 \sigma^2} \\
a &= \sqrt{2 \cdot \lambda_2} \cdot \left( \frac{\lambda_1 \sigma_x{}^2 + \sqrt{\lambda_1{}^2 \sigma_x{}^4 - 2\lambda_1 \sigma_x{}^2}}{2\lambda_1 \sigma^2} \right) \\
b &= 1 / \sqrt{2 \cdot \lambda_2}
\end{aligned}
$$

If $\lambda_1 \sigma_x{}^2 \gg 1$, these equations are approximated as next.

$$
\begin{aligned}
a \cdot b &\simeq \left( 1 - \frac{1}{2\lambda_1 \sigma_x{}^2} \right) \\
a &= \sqrt{2 \cdot \lambda_2} \cdot \left( 1 - \frac{1}{2\lambda_1 \sigma_x{}^2} \right) \\
b &= 1 / \sqrt{2 \cdot \lambda_2}
\end{aligned}
$$

Here, $a \cdot b$ is not equal to 1. That is, decoder is not a inverse function of encoder. In this case, the scale of latent space becomes slightly bent in order to minimize entropy function. As a result, good fitting of parametric PDF $P(z) \sim P_\psi(z)$ could be realized while proportional relationship $P(z) \propto P(x)$ is relaxed.

Next, the case of $h(d) = \log(d)$ is examined. By applying $h(d) = \log(d)$ and introducing a minute variable $\delta$, Eq. (66) can be expanded as follows.

$$
Loss = \log(a) + \log(\sigma_x \sqrt{2\pi e}) + \lambda_1 \cdot \log \left( (a \cdot b - 1)^2 + \delta \right) + \lambda_2 \cdot b^2 \quad (68)
$$

By solving $\frac{\partial Loss}{\partial a} = 0$ and $\frac{\partial Loss}{\partial b} = 0$ and setting $\delta \to 0$, $a$ and $b$ are derived as follows.

$$
\begin{aligned}
a \cdot b &= 1 \\
a &= \sqrt{2 \cdot \lambda_2} \\
b &= 1/\sqrt{2 \cdot \lambda_2}
\end{aligned}
\tag{69}
$$

Here, $a \cdot b$ is equal to 1 and decoder becomes a inverse function of encoder regardless of the variance $\sigma_x{}^2$. In this case, good proportional relation $P(z) \propto P(x)$ could be realized regardless of the fitting $P\psi(z)$ to $P(z)$.

Considering from these result, there could be a guideline to choose $h(d)$. If the parametric PDF $P\psi(\boldsymbol{z})$ has enough ability to fit the real distribution $P(\boldsymbol{z})$, $h(d) = \log(d)$ could be better. If not, $h(d) = d$ could be better.

# E   DETAIL OF THE EXPERIMENT IN SECTION 4.2

In this section, we provide further detail of experiment in section 5.2. First, we describe the detail of following four public datasets:

**KDDCUP99 (Lichman (2013))** The KDDCUP99 10 percent dataset from the UCI repository is a dataset for cyber-attack detection. This dataset consists of 494,021 instances and contains 34 continuous features and 7 categorical ones. We use one hot representation to encode the categorical features, and eventually obtain a dataset with features of 121 dimensions. Since the dataset contains only 20% of instances labeled -normal- and the rest labeled as -attacks-, -normal- instances are used as anomalies, since they are in a minority group.

**Thyroid (Lichman (2013))** This dataset contains 3,772 data sample with 6-dimensional feature from patients and can be divided in three classes: normal (not hypothyroid), hyperfunction, and subnormal functioning. We treat the hyperfunction class (2.5%) as an anomaly and rest two classes as normal.

**Arrhythmia (Lichman (2013))** This is dataset to detect cardiac arrhythmia containing 452 data sample with 274-dimensional feature. We treat minor classes (3, 4, 5, 7, 8, 9, 14, and 15, accounting for 15% of the total) as anomalies, and the others are treated as normal.

**KDDCUP-Rev (Lichman (2013))** To treat "normal" instances as majority in the KDDCUP dataset, we keep all "normal" instances and randomly pick up "attack" instances so that they compose 20% of the dataset. In the end, the number of instance is 121,597.

Next, hyper parameter for RaDOGAGA is described in table 2. First and second column is number of neuron. For DAGMM, we set same number of neuron in table 2 and $(\lambda_1, \lambda_2)$ as (0.1, 0.005). Optimization is done by Adam optimizer with learning rate $1 \times 10^{-4}$ for all dataset.

Table 2: Hyper parameter for RaDOGAGA

| Dataset | Autoencoder | EN | $\lambda_1(d)$ | $\lambda_2(d)$ | $\lambda_1((log(d)))$ | $\lambda_2(log(d))$ |
|---|---|---|---|---|---|---|
| KDDCup99 | 60, 30, 8, 30, 60 | 10, 4 | 100 | 1000 | 10 | 100 |
| Thyroid | 30, 24, 6, 24, 30 | 10, 2 | 100 | 10000 | 100 | 1000 |
| Arrhythmia | 10, 4, 10 | 10, 2 | 1000 | 100 | 1000 | 100 |
| KDDCup-rev | 60, 30, 8, 30, 60 | 10, 2 | 1000 | 100 | 100 | 100 |

In addition to experiment in main page, we also conducted experiment with same network size as in (Zong et al. (2018)) with parameters in table 3

Now, we provide results of setting in table 3. In table 4, RaDOGAGA- and DAGMM- are results of them and DAGMM is result cited from (Zong et al. (2018)). Even with this network size, our method has boost from baseline in all dataset.

Table 3: Hyper parameter for RaDOGAGA(referring (Zong et al. (2018)))

| Dataset | Autoencoder | EN | $\lambda_1(d)$ | $\lambda_2(d)$ | $\lambda_1((log(d))$ | $\lambda_2(log(d))$ |
|---------|-------------|-----|--------------|--------------|---------------------|---------------------|
| KDDCup99 | 60, 30, 1, 30, 60 | 10, 4 | 100 | 100 | 100 | 1000 |
| Thyroid | 12, 4, 1, 4, 12 | 10, 2 | 1000 | 10000 | 100 | 10000 |
| Arrhythmia | 10, 2, 10 | 10, 2 | 1000 | 100 | 1000 | 100 |
| KDDCup-rev | 60, 30, 1, 30, 60 | 10, 2 | 100 | 100 | 100 | 1000 |

Table 4: Average and standard deviations(in brackets) of Precision, Recall and F1

| Dataset | Methods | Precision | Recall | F1 |
|---------|---------|-----------|--------|-----|
| KDDCup | DAGMM | 0.9297 | 0.9442 | 0.9369 |
| | DAGMM- | 0.9338(0.0051) | 0.9484(0.0052) | 0.9410(0.0051) |
| | RaDOGAGA-(L2) | 0.9455(0.0016) | 0.9608(0.0018) | 0.9531(0.0017) |
| | RaDOGAGA-(log) | 0.9370(0.0024) | 0.9517(0.0025) | 0.9443(0.0024) |
| Thyroid | DAGMM | 0.4766 | 0.4834 | 0.4782 |
| | DAGMM- | 0.4635(0.1054) | 0.4837(0.1100) | 0.4734(0.1076) |
| | RaDOGAGA-(L2) | 0.5729(0.0449) | 0.5978(0.0469) | 0.5851(0.0459) |
| | RaDOGAGA-(log) | 0.5729(0.0398) | 0.5978(0.0415) | 0.5851(0.0406) |
| Arrythmia | DAGMM | 0.4909 | 0.5078 | 0.4983 |
| | DAGMM- | 0.4721(0.0451) | 0.4864(0.0464) | 0.4791(0.0457) |
| | RaDOGAGA-(L2) | 0.4897(0.0477) | 0.5045(0.0491) | 0.4970(0.0484) |
| | RaDOGAGA-(log) | 0.5044(0.0364) | 0.5197(0.0375) | 0.5119(0.0369) |
| KDDCup-rev | DAGMM* | 0.937 | 0.939 | 0.938 |
| | DAGMM- | 0.9491(0.0163) | 0.9498(0.0158) | 0.9494(0.0160) |
| | RaDOGAGA-(L2) | 0.9761(0.0057) | 0.9761(0.0056) | 0.9761(0.0057) |
| | RaDOGAGA-(log) | 0.9791(0.0036) | 0.9799(0.0035) | 0.9795(0.0036) |

## F  DETAIL OF THE EXPERIMENT 4.3

In this section, we provide further detail of experiment in section 5.3. For both RaDOGAGA and beta-VAE, we first extract feature with following Convolution Neural Network(CNN).

CNN(9, 9, 2, 64, GDN)-CNN(5, 5, 2, 64, GDN)-CNN(5, 5, 2, 64, GDN)-CNN(5, 5, 2, 64, GDN).

Here, CNN(w, h, s, c, f) is a CNN layer with kernel size (w, h), stride size s, dimension c, and activate function f. GDN(Ballé et al. (2015)) is often used in image compression. Then, we reshape feature map and send to autoencoder as follows.

FC(1024, 8192, softplus)-FC(8192, 256, None)-FC(256, 8192, softplus)-FC(256, 1024, softplus)

FC(i, o, f) is FC layer with input dimension i, output dimension o and activate function f. None means no activate function. Note that, for beta-VAE, since it produces mean and variance, the bottom of the encoder has 2 branches.

$(\lambda_1, \lambda_2)$ is as (1.0, 0.1) is set for RaDOGAGA and $\beta$ is set as $1 \times 10^{-4}$ for beta-VAE.

Optimization is done by Adam optimizer with learning rate $1 \times 10^{-4}$.

## G  PDF MATCHING WITH BETA-VAE

For the reader with interest, we provide the result of experiment in section 5.1 with beta-VAE. Network consists of FC layers of which have the same neuron numbers as the DAGMM and RaDO-GAGA. We set $\beta$ as 0.001. Figure 10a and 10b depict results. Since VAE dose not support Jacobian controlling, $Px(\boldsymbol{x})$ can not be mapped into $Pz_\psi(\boldsymbol{z})$ tidily.

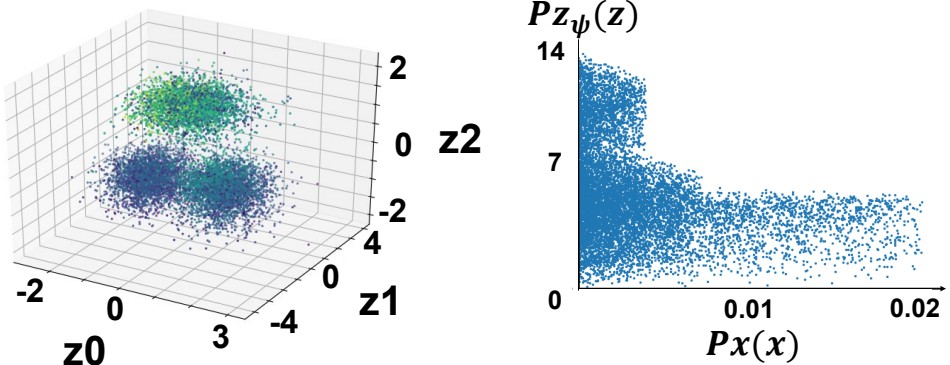

(a) Plot of $z$. Even though it captures three components of input data source $s$, PDF is quite different from that of $s$.

(b) Plot of of $Px(x)$ (x-axis) and $Pz_\psi(z)$ (y-axis). No clear correlation can been seen.

Figure 10: Result of PDF estimation with toy data (beta-VAE)

