# OpenReview forum: "RATE-DISTORTION OPTIMIZATION GUIDED AUTOENCODER FOR GENERATIVE APPROACH"
_ICLR.cc/2020/Conference — Reject_

### Official Review · AnonReviewer2 · 2019-10-16
**Official Blind Review #2**

**Rating:** 1

**Review:**

The paper propose a noisy autoencoder that considers the jacobian between data and latent spaces to match the corresponding densities. This idea has already been proposed elsewhere, and here it is applied to autoencoders. Overall I had hard time understanding the paper, the motivation, the main contribution or the claim, the model definition and the jacobian method. The paper is poorly written, with lots of issues in math notation and poor motivation and explication of what the sections are introducing, and what parts of the presentation is novel and what is already known. Lots of the math machinery is too vague to follow.

The distribution p(z) is unclear, and whether z is random variable or not. It seems that “z" is a non-random variable, and then adding noise \eps makes it stochastic. However, then p(z) without \eps does not make any sense since z is not random. It seems that p(z) is maybe a prior distribution instead (or maybe the variational posterior?), but then adding \eps noise to an already stochastic variable is strange. Overall I have hard time understanding the motivation of the two discrepancies in eq 4, what is the point of adding more noise to “z”? This seems some kind of noisy or perhaps robust AE variant, but the paper does not explicate this. I have hard time following the eqs 8-15. I am not convinced of the orthogonality argument, and I fail to see what this section tries to show or demonstrate. It seems that eq 14 is the final result, but its difficult to follow due to most terms in eq 14 being undefined. Optimizing eq 14 seems trivial since we can always match pz and pz_\varphi easily with neural networks, or similarly the two x-distributions.

In the experiment 4.1. the proposed method seems to achieve matching densities, although the distributions are wrongly normalized. How does the density matching improve? All three methods seem to have equally good scatters. The benchmarks on table 1 show clear improvement with the method. The face experiment is unconvincing since the VAE spreads variance across all latent dimensions while RADO seems to compress them to just first 20 or so. If one would visualise the z_100 there would be no variance in RADO and possibly some variance in VAE. The paper also should compare their model to simple MNIST/VAE to highlight what problems are there in standard approaches (such as VAE), and how does the proposed method alleviate them.

Overall the paper is poorly presented and difficult to follow. Despite this the method does seem to work remarkably well, and the Jacobian idea is clearly very promising. Nevertheless in its current form the paper is badly premature for ICLR, and needs a lot more work and polish to be made understandable for wider ML audience.

Minor comments
o Px(x), x1, x2 are probably missing subscripts
o The point of eq 5 is unclear, it seems unnecessary. It also does not contain h(), which is claimed after eq6
o The log pz(z) in eq 4 is not entropy
o eq 8 is unclear, is the dx a derivative, distance or change?
o the $^t$ prefix notation is confusing, what does it mean?
o what is the \sim and line notation in eq 5?
o what are the products in eq9, are these inner products?
o in eq 13 pz, pxd or hat(pxd) have not been introduced or defined


**Experience Assessment:**

I have read many papers in this area.

**Review Assessment: Checking Correctness Of Derivations And Theory:**

I assessed the sensibility of the derivations and theory.

**Review Assessment: Checking Correctness Of Experiments:**

I assessed the sensibility of the experiments.

**Review Assessment: Thoroughness In Paper Reading:**

I read the paper at least twice and used my best judgement in assessing the paper.

---

> ### Author Response · Authors · 2019-11-15
> **Response to the comments (part 2)**
>
> >>Regarding experimental results
> >>In the experiment 4.1. the proposed method seems to achieve matching densities, although the distributions are wrongly normalized. How does the density matching improve? All three methods seem to have equally good scatters.
>
> As written in the result section, even though the baseline method (DAGMM) also captured good scatter, the density is not estimated adequately. Figure 5 (in revised version) depicts plot of Px(x) (x-axis) and Pzψ(z) (y-axis).  It is obvious that we can see the proportionality between Px(x) and Pzψ(z), while we cannot see the tendency in the baseline. This is also quantitatively evaluated. The correlation coefficients are 0.691 (baseline) vs 0.997 or more (ours). (Originally we showed residual of linear regression though, for intuitive understanding, we replaced it by correlation coefficients.)
> For the easy following, we also updated the caption and description in the main text.
>
> >>The face experiment is unconvincing since the VAE spreads variance across all latent dimensions while RADO seems to compress them to just first 20 or so. If one would visualize the z_100 there would be no variance in RADO and possibly some variance in VAE.
>
> Yes, what you said happens. This means that our model correctly works as PCA as in theory. The point is the variance of latent variables directly correlated
> This is the experiment to confirm that latent variables of our model work as PCA components. To make it further explicit, we added this statement at the beginning of the section.
>
> Let’s say if we want to find an important latent variable in terms of the influence on the metrics function, what should we do?
> In our model, we can find that latent variable easily, since the variance of z is directly related to the visual. In (beta-)VAE, as we described, variance and impact to visual are uncorrelated. Thus, for example, we need to visualize all latent variables to find the important ones. Moreover, even if you come up with you run PCA for the latent variables of VAE, you need to set the number of PCA components and may struggle to decide how many components are appropriate. In our model, it is automatically optimized in terms of minimizing entropy.
> Consequently, the latent variable in our model is quantitatively understandable. We believe this character is helpful for the interpretation of latent variables and meta-prior of data.
>
> >>The paper also should compare their model to simple MNIST/VAE to highlight what problems are there in standard approaches (such as VAE), and how does the proposed method alleviate them.
>
> An Experiment with MNIST could be worth-doing though, we demonstrated the above characteristic clearly with CelebA dataset.
>
> In terms of the interpretation of latent variables, some of the standard approaches are visualizing or evaluate independencies of variables as in  (Lopez et al., 2018; Chen et al., 2018; Kim & Mnih, 2018; Chen et al., 2016). They do not directly evaluate the importance of latent variables on the metric function(such as MSE or SSIM). In our method, this can be quantitatively measured like PCA.
> Note that, we do not intend to claim this way of analysis is always better than previous ways. We argue that making use of PCA like analysis as an option and incorporating them will promote further interpretation of latent variables.
>
> >>For minor comments
> o The point of eq 5 is unclear, it seems unnecessary. It also does not contain h(), which is claimed after eq6
> Equation 5 (6 in the revised version) is a condition for function D(・,・). As long as D(・,・) can be approximated as eq 5, it can be applied. We added this explanation.
>
> o The log pz(z) in eq 4 is not entropy
> We fixed it.
>
> o eq 8 is unclear, is the dx a derivative, distance or change?
> It is derivative. We added the notation.
>
> o the t prefix notation is confusing, what does it mean?
> The t denotes the transpose of a matrix. We added the notation.
>
> o what is the \sim and line notation in eq 5
> \simeq denotes approximation.
>
>  o what are the products in eq9, are these inner products?
> It is a multiplication. We removed dots.
>
> oi n eq 13 pz, pxd or hat(pxd) have not been introduced or defined
> hat(pxd) is defined as “let hat(pxd) be estimated probability of xd.”　We added definition pz, pxd as the true PDF of z and xd

---

> ### Author Response · Authors · 2019-11-15
> **Response to the comments (part 1)**
>
> Thank you for your time and comments. Thanks to your comments, we could improve the paper a lot! We hope the revised version and these rebuttal comments will solve your confusion. Please understand that we revised to some extent in order to respectfully deal with your comments and to make our claim more persuasive.
>
> >>Regarding our motivation and connection to prior works.
> Because we have a background not only about deep autoencoder but also about image compression, we have overlooked a gap between image compression and VAE as you pointed out. To make our motivation and the difference from previous work clear, we added section 3. Figures 1 and 2 describe the overview. Please find the following points are described.
>
> Our method is based on rate-distortion optimization(RDO) of transform coding for image compression. RDO is a method to improve quality in image compression with orthonormal transform coding.
> (https://en.wikipedia.org/wiki/Rate-distortion_optimization)
> For the readers who are not familiar with RDO, we added the overview of RDO, its connection to VAE, and our motivation to introduce RDO to autoencoder (not VAE).
>
> Here is a summary of the added section.
> According to RDO theory, the condition of optimization in transform coding is that: (i) transform data using orthonormal basis (orthogonal is not enough) such as DCT, KLT, and so on (ii) quantize by uniform quantizer for all channels which cause uniform noise (iii) assign the optimum entropy code.
> Our intuition is that if the equivalent noise is added to latent variables z and rate-distortion is optimized, z should have orthonormality. Consequently, Jacobian becomes constant automatically.
> Accordingly, z is obtained deterministically. The reason to add equivalent noise is based on this idea. Since our model minimizes entropy of z, there is no prior for p(z).
>
> Actually, RDO can be analogously discussed in VAE and there are works considering rate-distortion trade-off into VAE. But in the way to assume fixed distribution as prior like VAE, even if rate-distortion is optimized, orthonormality is not guaranteed, and Jacobian is not constant.
>
> As we mentioned in section 2., Flow-based models take Jacobian of into account (we assume this model would be ‘elsewhere’). Although, in Flow method, the encoder and the decoder need to be a bijection, which means the dimension of the data space and the latent space is the same. On the other hand, our model can compress data into a lower dimension with a constant Jacobian.
>
> >> Regarding theory part
> >>It seems that eq 14 is the final result
> The final cost function is eq. (5) and it is substantially the same as eq. (14) ((15) in revised version). This equation is related to rather the needlessness of ELBO than orthonormality.
> To avoid confusion, we split the section into the method part and theoretical part. Also, we enhanced the purpose of each equation.
>
> >>orthogonal argument
> Although we gave full proof in Appendix A,  in the main text, we rely on  Rolinek et al., (2019) for the argument of orthogonality since it is already proved. We show the proof of constant Jacobian and orthonormality, combining the orthogonality.
>
> >>Optimizing eq 14 seems trivial since we can always match Pz and Pz_\varphi easily with neural networks, or similarly the two x-distributions.
>
> You seem to imagine matching the KL-divergence of Pz as prior and pz\varphi or something like that. As it is mentioned, Jacobian J is not constant in the most previous methods. Thus, even if just Pz and Pz_\varphi were matched, it does not mean estimating p(x), which is our goal, is achieved.

---

### Official Review · AnonReviewer3 · 2019-10-24
**Official Blind Review #3**

**Rating:** 3

**Review:**

This paper aims to obtain latent representation of data such that probability density for the real space can be calculated correctly from that in the latent space. The authors optimize a loss function that has components related to parametric probabilistic distribution and auto encoder simultaneously. While this might be an important problem (I am not sure), the paper is not written and organized well which makes a through evaluation very difficult. I provide below some of the problems with this the paper:

Why the introduced method is better than VAE as a generative model for capturing the latest representation is not explained well. It is not also used as a baseline in most of the experiments.

The motivation for having the third term in Equation (4) needs to be explained. Also what is h() in the second term. The authors only describe briefly both terms together after they used it here but failed to describe what each term is.  Why there is an h for the second term but not for the third term. h() becomes more clear much later in the paper but when it is used the first time, it not defined.

I believe A in Equation (5) should be also positive-definite.

What is L(x) in Equation (8).  It needs to be defined.


Experiments:
1-	It is useful to also plot the original data in space s to see how the results in Figure 2 make sense.
2-	Figure 3 is not clear.
3-	In the Anomaly detection experiments, the authors make two assumptions that usually do not exist in real-worlds: (1) they assume that they have access to training set that only contains normal cases. (2) They assume that they know the correct rate of anomaly. I think both these assumptions are very restrictive and unreal. While these assumptions are used for all the comparing methods, it is not obvious how different algorithms behave in real scenario.
4-	Figure 4 and what it represents is not clear.

Writing Problems:
1-	In the text of paragraph before Figure 1, Eq. (5) in “in the second term of Eq. (5)” is a typo and should be Eq. (4).
2-	In the paragraph before Figure 1, the following sentence is not complete: “Then, averaging Eq. (4) according to distribution, x~P_x(x) and epsilon~ P(epsilon).”
3-	Section 4.2.1: “there is a difference is PDF → “there is a difference in PDF”

**Experience Assessment:**

I have published one or two papers in this area.

**Review Assessment: Checking Correctness Of Derivations And Theory:**

I assessed the sensibility of the derivations and theory.

**Review Assessment: Checking Correctness Of Experiments:**

I assessed the sensibility of the experiments.

**Review Assessment: Thoroughness In Paper Reading:**

I read the paper thoroughly.

---

> ### Author Response · Authors · 2019-11-15
> **Response to the comments (part 2)**
>
> Regarding experiments:
> >>1- It is useful to also plot the original data in space s to see how the results in Figure 2 make sense.
> Thanks for your point. We added the plot of the original data source.
>
> >>2- Figure 3 is not clear.
> Figure 3 (Figure 5 in revised version) depicts plot of Px(x) (x-axis) and Pz_\psi (z) (y-axis). A linear plot means that the probability density of Px(x) is tidily mapped into the latent space. Thanks to this property, Px(x) can be estimated by Pz_\psi (z) in our model. It is obvious that DAGMM does not have this trait.  This is also quantitatively evaluated. The correlation coefficients are 0.691 (baseline) vs 0.997 or more (ours).
> For the easy following, we revised the caption and description in the main text.
>
> >>3- In the Anomaly detection experiments, the authors make two assumptions that usually do not exist in real-worlds: (1) they assume that they have access to a training set that only contains normal cases. (2) They assume that they know the correct rate of anomaly. I think both these assumptions are very restrictive and unreal. While these assumptions are used for all the comparing methods, it is not obvious how different algorithms behave in a real scenario.
>
> I understand that there is an unrealistic assumption, but this setting is established and widely admitted in this anomaly detection task.
> Regardless of this assumption is realistic or not, density estimation remains a critical issue, and better estimation provides better performance. Although investigating the performance in a truly real scenario might be future work, we argue this point is not a defect to show the validity of our method.
>
> >>4- Figure 4 and what it represents is not clear.
> This is caused because we could not tell you the purpose of this experiment sufficiently. This is an experiment to show an important property of our model: our model behaves as PCA, where the energy of acquired latent space is concentrated on several principal components and the influence of each component can be evaluated quantitatively
> The two on the left of Fig. 4 (Fig. 6 in the revised version) is the variance of the latent space. Since our model works as PCA, the variance is concentrated in a few dimensions. Two on the right shows that the influence of minute displacement of each z to the real image is the almost constant in our model while it is varied in beta-VAE. Thus, we can evaluate the importance of latent variables by variance like PCA. We added the caption and enhanced the purpose of the experiment.
>
> >>Regarding writing issue
> Thank you for pointing. We fixed them.

---

> ### Author Response · Authors · 2019-11-15
> **Response to the comments (part 1)**
>
> Thank you for your time and valuable comments. Please understand that we revised to some extent in order to respectfully deal with your comments and to make our claim more persuasive.
>
> First of all, we added an explicit discussion about our motivation, idea, and connection among prior works which we overlooked before. Figures 1 and 2 give an overview.
>
> >>Why the introduced method is better than VAE as a generative model for capturing the latest representation is not explained well. It is not also used as a baseline in most of the experiments.
>
> We added section 3 to make the relation and difference between VAE and our method much explicit.
> By this section, we believe the following two points stated repeatedly in the entire text became easy to follow. These are difference not only from VAE but also from other autoencoders without Jacobian control. Since you seem not to care about the second point very much, we would appreciate your attention to it. We insist the second point promotes the interpretation of latent variables which has been discussed as one of the most important problems of deep learning.
>
> (i) the probability distribution of the latent space obtained by this model is proportional to the
> probability distribution of the real space because Jacobian between two spaces is constant;
> (ii) our model behaves as non-linear PCA, where the energy of acquired latent space is concentrated on several principal components and the influence of each component can be evaluated quantitatively
>
> The experiment in section 5.1(in the revised version) demonstrates the first feature. Furthermore, an experiment in section 5.2 shows the validity of the practical task. The second feature is examined in the experiment in section 5.3. Figure. 4 (Fig. 6 in revised ver.) demonstrates the second property.
>
> DAGMM is known as a model to estimate the density better than (beta-)VAE and suitable for baseline though, we added the result of the experiment in section 5.1 (toy data task) in Appendix G. As it is mentioned before, in VAE, Jacobian is not constant and Px(x) and Pz_\psi(z) have no correlation. We can also move this to the main text if it is necessary.
>
> In the anomaly detection task, we added the score of VAE cited from Liao et al. (2018). Actually GMVAE is also a VAE based method. Unfortunately, we couldn’t reproduce the result by ourselves though, our model performs significantly better compared with that. Since GMVAE does not care about Jacobian and maximizing ELBO as well, it essentially includes disorder in the density estimation.
>
> >>The motivation for having the third term in Equation (4) needs to be explained. Also, what is h() in the second term? The authors only describe briefly both terms together after they used it here but failed to describe what each term is.  Why there is an h for the second term but not for the third term. h() becomes more clear much later in the paper but when it is used the first time, it not defined.
>
> We added the explanation the third term and h() when it is used the first time. The second and third terms are actually decomposition of D(x, x_\breve) as shown in Rolinek et al., 2019. By this decomposition, we can independently control the reconstruction loss and scaling Jacobian and lead to better performance.
>
> >> I believe A in Equation (5) should be also positive-definite.
> Yes, it is. We added the description.
>
> >>What is L(x) in Equation (8). It needs to be defined.
> We added the definition of L(x).

---

### Official Review · AnonReviewer4 · 2019-11-04
**Official Blind Review #4**

**Rating:** 3

**Review:**

Summary of the paper: The authors propose a latent variable model RaDOGAGA, a generative autoencoding model. The model is trained via a tradeoff between distortion (the reconstruction error) and the rate (the capacity of the latent space, measured by entropy). The paper provides an analysis of theoretical properties of their approach, and presents supporting experimental results.

Review tl;dr: weak reject, for three main reasons:
(i) While the existing literature around VAEs, beta-VAEs,  and Rate-Distortion theory is mentioned in the related work, the connections are not nearly discussed sufficiently.
(ii) On top of (i), the derivation of their loss function and architecture is not sufficiently motivated. This is in astonishing contrast to 1.5 pages of main text and 8 pages of (much appreciated!) analysis of properties.
(iii) Given the paper is clearly related to existing approaches in the literature, the experiments would require a much more careful comparison to existing models. It remains unclear why an interested user should favor your model over conceptually simpler generative models with fewer hyperparameters.

Detailed review:

Nota bene: This review is a late reassignment. While I reviewed the paper to the best of my ability, time constraints did not allow me to review parts of the paper in depth.  I am open to reassess my review during the second stage.

Connection to prior art: As a probabilistic, neural autoencoding model, the connections to the family of VAE models are obvious. The loss function (eq. (4)) still looks very much like the ELBO, where the typical conditional log-likelihood was split into two distortion terms. How is this different from e.g. a beta-VAE? Particularly, what is the connection between the rate-distortion analysis of beta-VAE by Alemi et al. and yours? These things need to be discussed explicitly, with more than a sentence or two in the related work section.
A lesser, but still important omission in your discussion of prior work: The Jacobian of the generator has also been studied, even for the VAE, cf. e.g. [1]. I believe this deserves more attention in your assessment of prior art.

Motivation: You use two distortion terms: actual sample vs. undistorted reconstruction. Why is that? What is the interpretation of the multipliers? How do I choose them? Why is a large part of your architecture (the pipeline from x to \hat(x)) actually deterministic? Why are you using the entropy of the prior over the latents, rather than the KL divergence between encoder and a prior? I think an interested reader could learn much more from your paper if you discussed your model embedded in th related work rather than in isolation.

Theory: Due to aforementioned time constraints, I was not able to review the extensive theoretical analysis in depth. Still, I would strongly recommend structuring the respective sections more clearly. Separate model and architecture description from the theoretical analysis; precisely formulate your claims. In particular, state your assumptions clearly. For instance, you assume "that each function's parameter is rich enough to fit ideally" (and similar e.g. in Appendix A). Does this only mean that the true distributions are part of the parametric family? What if this is not the case? Do your parameters need to be in the optimum for your analysis to hold true?

Given that the full 20-page manuscript spends 10 pages on theory, I think this contribution is not given appropriate space in the main text.

Experiments: There are three experiments: a simple 3D proof of concept; anomaly detection; analysis of the latent state in CelebA. As mentioned in my review of the methods section, I believe the approach to be very similar to established models. None of the experiments provides convincing evidence why I should prefer the new, arguably more complex model.
For instance, I would have much preferred that you investigate properties of your model against alternatives over the anomaly detection experiments, which did not further my understanding of the proposed model.

Summary: The paper tackles an important problem, namely the lack of control over the latent embedding in autoencoding generative models. I believe the author's contribution can be valuable, and I particularly appreciate the effort to investigate theoretical properties. As is, the case is not sufficiently convincing to be accepted, but I encourage the authors to improve the paper.

Minor comments:
1. While I appreciate a pun, I would recommend to rename the model along with the acronym to a more concise name.
2. Please revise your notation and typsetting. Examples: x1 instead of x_1, f of f(\cdot) instead of f(), \log instead of log.
3. Introduce acronyms before using them (e.g. VAE, MSE, SSIM), even when they seem obvious to you.
4. Please carefully check the manuscript for typos, missing articles, missing spaces etc.
5. Your citations are inconsistent, in that they sometimes use first names, sometimes first name initials, and sometimes no first names.
6. To my knowledge, the term scale function does not have an obvious definition. I think you are simply referring to monotonically increasing functions. Please clarify!
7. Your figures should be understandable without too much context, they need more detailed captions.

[1] http://proceedings.mlr.press/v84/chen18e.html

**Experience Assessment:**

I have published one or two papers in this area.

**Review Assessment: Checking Correctness Of Derivations And Theory:**

I assessed the sensibility of the derivations and theory.

**Review Assessment: Checking Correctness Of Experiments:**

I assessed the sensibility of the experiments.

**Review Assessment: Thoroughness In Paper Reading:**

I read the paper at least twice and used my best judgement in assessing the paper.

---

> ### Author Response · Authors · 2019-11-15
> **Response to the comments (part 2)**
>
> >>Regarding the experimental result.
> First of all, actually, our model does not increase model complexity even though you concerned about this point.
> When we compared with our model and DAGMM (baseline model), the number of network parameters is completely the same. We added this point explicitly.
> Nevertheless, our model provides a significant performance boost in the anomaly detection task.
>
> Experiment with toy data is executed to confirm our model’s property though, it also supports the result of anomaly detection. In DAGMM, the relation PDF of x and z is unclear. On the other hand, in our model, the PDFs of x and z are close to proportional. That means, our model can capture the probability of real data methodically in the latent space. This fact should be very intuitive to explain the performance boost in the anomaly detection task in which PDF estimation is a critical issue. Other comparison methods also essentially lead the disorder in the density estimation like DAGMM because the Jacobian is not controlled.
>
> In the analysis of the latent state in CelebA, we assume that since we could not tell you the purpose of the experiment enough, it was not convincing for you.
> This is an experiment to confirm that the latent variable in our model works as PCA components, and the influence of each component can be evaluated quantitatively as in theory while (beta-)VAE does not have this property. We revised this sections and captions for easy following.
> The two on the right of Fig. 6 in the revised version show the scaling between the latent and metric dependent data space. (c) shows the scaling in VAE is anisometric, and (d) shows the scaling in ours is isometric.
> The two on the left of Fig. 6 in the revised version is the variance of the latent space. Since the scaling of z in our model is isometric, the variance shows the importance of each latent variable like PCA.
>
> Consequently, we believe our experimental results demonstrate the validity of our method decently.
>
> PCA can simultaneously disentangle the data and estimate the importance of latent variables by variance. We believe this trait is very helpful to the interpretation of the latent variable of deep models.
>
>
> >>minor issues
> We fixed the minor issues you pointed (we would appreciate if you could be indulgent of a bit long model acronym).
> We also promise to request a grammatical check by a native no later than the camera-ready version.

---

> ### Author Response · Authors · 2019-11-15
> **Response to the comments (part 1)**
>
> Thank you for your time and valuable comments.
> From your comments, we found that our work would be closely related to a practical method of isometric embedding of Riemannian manifold.
> Because our background is not only deep autoencoders but also image compression, we have overlooked that there is a gap between image compression and VAE.
> Please understand that we revised to some extent in order to respectfully deal with your comments and to make our claim more persuasive.
>
> >>Regarding our motivation and its connection with prior works
> We added an explicit discussion about our motivation, idea, and connection among prior works. Figures 1 and 2 give an overview. Please find the following discussion are added.
> The term “Rate-distortion optimization” or “RDO” is a method to improve quality in image compression with orthonormal transform coding.
> https://en.wikipedia.org/wiki/Rate-distortion_optimization
> Prior works of deep image compression such as Balle et al., 2018 also used RDO. We added an overview of RDO and our motivation.  The derivation of our idea is based on the analogy of orthonormal transform coding.
> We also added the analogy and difference between VAE and our idea. The summary is as follows.
>
> According to RDO theory, the condition of optimization in transform coding is that: (i) transform data deterministically using orthonormal basis (orthogonal is not enough) such as DCT, KLT, and so on (ii) quantize by uniform quantizer for all channels which cause uniform noise (iii) assign the optimum entropy code.
> Our intuition is that if the equivalent noise is added to latent variables z and rate-distortion is optimized, z should have orthonormality. Consequently, Jacobian becomes constant automatically.
> Obeying this flow, z is obtained deterministically and the entropy is used rather than KL divergence between an encoder and a prior.
>
> Rate-distortion optimization condition for VAE and our model is contrasted as follows. In VAE, because PDF is fixed as prior, noise should be variable and scaling between data and latent spaces is also variable, meaning Jacobian is inconstant. In ours, because noise is uniform, PDF should be variable(parametric) and scaling is constant.  Thus, in our model, there is not fixed prior.
>
> About the loss function, The second and third terms in eq (5) are an approximate decomposition of D(x, x_\breve) as shown in Rolinek et al., 2019. By this decomposition, we can independently control the reconstruction loss and scaling Jacobian and lead to better performance.
>
> >> Relation to [1] http://proceedings.mlr.press/v84/chen18e.html
> Thank you for introducing an interesting paper to us. Thanks to your comment, we found that our work, especially Eqs. (10) and (11), would be related to an isometric embedding of Riemannian manifold where A(x) is a Riemannian tensor.  In this paper, the authors discussed the distance between two points is the shortest path on a Riemannian manifold induced by the transformation. Then, the impact on the domain data caused by the variance of latent variables is measured. This is related to the discussion of Fig. 6 (c) and (d). While VAE needs to find a winding road in the latent space that corresponds to the shortest path, in our model, a linear path in the latent space expected to be connected to that.  While we did not include this discussion this time because of page limitation though, this will be our future work.
>
>
> >> Do your parameters need to be in the optimum for your analysis to hold true?
> Strictly speaking, yes. Although, as experiment result showed, when parameters are optimized decently, it works almost as in theory even though there remains the left behind margin.

---

### Decision · Program_Chairs · 2019-12-19

**Decision:**

Reject

**Comment:**

Agreement by the reviewers: although the idea is good, the paper is very hard to read and not accurately enough formulated to merit publication.

This can be repaired, and the authors should try again after a thorough revision and rewrite.